# Transdifferentiation of plasmatocytes to crystal cells in the lymph gland of *Drosophila melanogaster*

Julien Marcetteau[1,5], Patrícia Duarte[1,5], Alexandre B Leitão [ID] [2✉] & Élio Sucena [ID] [1,3,4✉]

## Abstract

Under homeostatic conditions, haematopoiesis in *Drosophila* larvae occurs in the lymph gland and sessile haemocyte clusters to produce two functionally and morphologically different cells: plasmatocytes and crystal cells. It is well-established that in the lymph gland both cell types stem from a binary decision of the medullary prohaemocyte precursors. However, in sessile clusters and dorsal vessel, crystal cells have been shown to originate from the transdifferentiation of plasmatocytes in a Notch/Serrate-dependent manner. We show that transdifferentiation occurs also in the lymph gland. In vivo phagocytosis assays confirm that cortical plasmatocytes are functionally differentiated phagocytic cells. We uncover a double-positive population in the cortical zone that lineage-tracing and long-term live imaging experiments show will differentiate into crystal cells. The reduction of Notch levels within the lymph gland plasmatocyte population reduces crystal cell number. This extension of a transdifferentiation mechanism reinforces the growing role of haematopoietic plasticity in maintaining homeostasis in *Drosophila* and vertebrate systems. Future work should test the regulation and relative contribution of these two processes under different immunological and/or metabolic conditions.

**Keywords** Haematopoiesis; Cell Plasticity; Crystal Cell; Haemocyte
**Subject Categories** Development; Immunology

## Introduction

Under homeostatic conditions, haematopoiesis in *Drosophila melanogaster* produces two functionally distinct, mature haemocyte classes: plasmatocytes and crystal cells (Rizki, 1957). Plasmatocytes are macrophage-like cells and makeup approximately 90–95% of the haemocyte population whilst crystal cells are larger, post-mitotic cells which produce prophenoloxidase and constitute the remaining 5–10% (Rizki, 1957, 1959; Lanot et al, 2001; Evans et al, 2003; Banerjee et al, 2019). These two cell populations have been typically distinguished based on their morphology (Rizki, 1957;

Shrestha and Gateff, 1982; Lanot et al, 2001) and through the expression of a range of specific markers and corresponding drivers (Lebestky et al, 2000; Kurucz et al, 2007; Evans et al, 2014). Several recent studies revealed that this classic classification may be overly simplistic, as single-cell transcriptomics and functional studies revealed that haemocytes cluster into a larger number of categories than originally thought (Cattenoz et al, 2020; Cho et al, 2020; Fu, 2020; Leitão et al, 2020; Tattikota et al, 2020; Girard et al, 2021; Coates et al, 2021; preprint: Brooks et al, 2024). Nonetheless, this knowledge does not contradict, but rather confirms, the notion that mature plasmatocytes express diagnostic markers such as the phagocytosis receptors Eater and NimrodC1 (Kocks et al, 2005; Kurucz et al, 2007; Hultmark and Andó 2022). The same applies to crystal cells, which can be identified by their expression of the RUNX transcription factor Lozenge (Lebestky et al, 2000) and activity driven by the crystal cell-specific enhancer, BcF6 (Tokusumi et al, 2009).

The first stage of haematopoiesis takes place in embryonic development, during which a population of cells of the pro-cephalic mesoderm adopts haemocyte fate (Holz et al, 2003). Haemocytes of this population subsequently migrate into the hemocoel (Tepass et al, 1994), into the haematopoietic pockets and into dorsal vessel-associated clusters (Makhijani et al, 2011). These cells persist throughout larval life and into adulthood (Holz et al, 2003; Wood and Jacinto, 2007; Letourneau et al, 2016). Also, during embryogenesis, lymph gland development is initiated from the cardiogenic mesoderm as a series of bilaterally symmetrical lobes, which flank the anterior end of the dorsal vessel (Rugendorff et al, 1994; Grigorian and Hartenstein, 2013). Throughout larval life, a second stage of haematopoiesis will take place in the lymph gland, primarily in the most anterior primary lobes. These comprise a medullary zone containing undifferentiated prohaemocytes surrounded by a cortical zone of maturing and mature haemocytes (Shrestha and Gateff, 1982; Leitão and Sucena, 2015, Jung et al, 2005; Crozatier and Meister, 2007; Blanco-Obregon et al, 2020; Kharrat et al, 2022). In addition, haematopoiesis occurs in the segmentally organized, sub-epidermal sessile haemocyte clusters of the larva (Márkus et al, 2009; Makhijani et al, 2011; Bretscher et al, 2015; Gold and Brückner, 2015; Leitão and Sucena, 2015) and, arguably, in the adult (see (Ghosh et al, 2015) versus (Sanchez Bosch et al, 2019)).

The classic haematopoiesis model establishes that the fate of prohaemocytes is defined through a binary decision between plasmatocyte and crystal cell differentiation programs

[1]Instituto Gulbenkian de Ciência, Rua da Quinta Grande 6, 2780-156 Oeiras, Portugal. [2]Fundação Champalimaud, Lisboa, Portugal. [3]Departamento de Biologia Animal, Faculdade de Ciências, Universidade de Lisboa, Edifício C2, Campo Grande, 1749-016 Lisbon, Portugal. [4]cE3c: Centre for Ecology, Evolution and Environmental Changes, Faculdade de Ciências, Universidade de Lisboa, Campo Grande, 1749-016 Lisbon, Portugal. [5]These authors contributed equally: Julien Marcetteau, Patrícia Duarte. ✉E-mail: alexandre.leitao@research.fchampalimaud.org; jesucena@fc.ul.pt

(Lebestky et al, 2000; Duvic et al, 2002; Krzemien et al, 2010a; Banerjee et al, 2019). This happens as early as during early second instar larvae (L2) when the cortical zone matures and involving intermediate cell types (Kocks et al, 2005; Sinenko et al, 2009; Krzemien et al, 2010a, 2010b; Ferguson and Martinez-Agosto, 2014; Blanco-Obregon et al, 2020; Spratford et al, 2021). Strong evidence has accumulated to sustain that Notch pathway activation, through its ligand Serrate, is necessary for crystal cell differentiation. This process appears to be restricted to the cortical zone of the lymph gland (Duvic et al, 2002; Lebestky et al, 2003; Jung et al, 2005; Blanco-Obregon et al, 2020) and the sessile haemocyte clusters (Leitão and Sucena, 2015). Interestingly, in the sessile clusters, crystal cells arise de novo through the Notch-dependent transdifferentiation of mature plasmatocytes (Leitão and Sucena, 2015), a finding that does not fit with the original "binary decision model" of crystal cell differentiation from a haemocyte precursor. More recently, this alternative mechanism of crystal cell production has been shown to also occur in the dorsal vessel (Cevik et al, 2019), and to possibly depend on oxygen-sensing neurons in sessile clusters that surround the caudal sensory cones (preprint: Corcoran et al, 2020). Importantly, this transdifferentiation process is also established for lamellocyte production, as shown by the Hultmark and Andó laboratories (Honti et al, 2010; Anderl et al, 2016; Csordás et al, 2021) and further reinforced in recent studies (Cho et al, 2020; Tattikota et al, 2020). Indeed, scRNA-seq data has shown that haemocyte populations, once subclustered and ordered along a pseudotemporal scale, exhibit lineages ranging from self-cycling plasmatocytes to terminal fate crystal cells fate or lamellocytes. More importantly, single-cell transcriptomic data related exclusively to the lymph gland suggests the existence of a plasmatocyte subpopulation with the potential to transdifferentiate into crystal cells, within this organ (Cho et al, 2020; Girard et al, 2021).

In light of this accumulated evidence in sessile clusters and dorsal vessel for a Notch-dependent plasmatocyte-to-crystal cell transdifferentiation process, we tested for its extension to the lymph gland, the main *Drosophila* haematopoietic organ.

## Results

### Cortical plasmatocytes transdifferentiate into crystal cells in a Notch-dependent manner

Notch activation is necessary and sufficient for crystal cell differentiation in the lymph gland (Lebestky et al, 2000, 2003; Duvic et al, 2002; Small et al, 2014) and necessary in the sessile clusters for the transdifferentiation of mature plasmatocytes into crystal cell (Leitão and Sucena, 2015). If transdifferentiation is taking place also in the lymph gland, several predictions ensue. First, Notch activation should be restricted to the cortical zone, which is to say, detectable only in cortical plasmatocytes; second, intermediate-stage cells should be found, i.e., expressing terminal differentiation markers; lastly, interfering with Notch activity in cortical plasmatocytes should reduce crystal cell numbers. In order to test these predictions, we took advantage of the Notch responsive element reporter line (NRE-eGFP).

First, combining NRE-eGFP with HmlΔ-DsRed (a cell marker that starts its expression in maturing haemocytes) and observing 12

dissected lymph glands could not reveal NRE-eGFP expression in the medullary zone, which hosts undifferentiated haemocytes (Fig. EV1A,a′). The restricted expression pattern of Notch activity to the intermediate and cortical zone of Hml+ haemocytes is consistent with crystal cell differentiation not occurring in the medullary zone.

A transdifferentiation model predicts the existence of a transient cell type, which simultaneously, or sequentially, expresses mature plasmatocyte and Notch activity markers. We stained lymph glands from L3 larvae carrying the NRE-eGFP reporter with the P1 antibody, specific against the Nimrod membrane protein expressed in differentiated phagocytically active plasmatocytes (Vilmos, 2004; Kurucz et al, 2007; Terriente-Felix et al, 2013) confirming the existence of a qualitatively distinct population of plasmatocytes activated for Notch in the cortical zone of the lymph glands (Fig. 1A,1a'). The predicted transient nature of this cell type could be verified further by quantifying each of the three cell populations expected under a transdifferentiation model: plasmatocytes (P1+ NRE−), committed crystal cells (P1− NRE+) and transdifferentiating plasmatocytes (P1+ NRE+) (Fig. 1B). As predicted, the vast majority (circa 80%) of the cortical zone is populated by plasmatocytes and, in accordance with the model, a double-positive population is comparatively small (Fig. 1B). Another, independent, criterion to further establish the qualitative differences between these three populations is cell size—a strong predictor of cell type in this system (Shrestha and Gateff, 1982; Leitão and Sucena, 2015). Indeed, cell size was found to differ significantly between cell populations (one-way ANOVA, $F = 173.3$, $P < 0.01$), with single P1+ cells being the smallest and single NRE+ the largest (Tukey HSD, diff = 21.89, $P < 0.01$); more importantly, the transdifferentiating cells show a distinct and intermediate size range (Tukey HSD, single P1+ vs P1+NRE+, diff = 13.96, $P < 0.01$ and single NRE+ vs P1+NRE+, diff = 7.93, $P < 0.01$) (Fig. 1C).

Having confirmed the existence of a Notch-activated transient stage, we proceeded to test if intermediate-stage cells, simultaneously expressing crystal cells and plasmatocytes terminal differentiation markers, could be observed. We stained lymph glands expressing a GFP-tagged Lozenge protein, a marker of committed crystal cells (Lebestky et al, 2000), with P1 antibody. As predicted, we observed a population of cells expressing both markers (Fig. 2A,a′). This result was further corroborated using Black-cell (BcF6-GFP), an earlier crystal cell marker that partially reproduces the endogenous pattern of the prophenoloxidase gene, PPO1 (Gajewski et al, 2007) and a Lozenge driver that mimics crystal cell expression of this transcription factor (Terriente-Felix et al, 2013) (Fig. EV1B–b',C–c').

We tested this hypothesis further by assessing whether crystal cells would descend from cortical plasmatocytes. Though expectedly rare, a transitional state consisting of triple-positive cells for plasmatocyte, Notch activation and crystal cell markers should exist within the cortical zone of third-instar lymph glands undergoing transdifferentiation. To explore this possibility, we used a BcF6-GFP reporter (Gajewski et al, 2007) that is activated by Lz. In hemizygous flies for the null allele LzS that lack crystal cells and phenoloxidase activity (Rizki and Rizki, 1981), this reporter will not be expressed but will display an expression pattern comparable to that of the endogenous PPO1 gene in flies wild-type for Lozenge function (Tokusumi et al, 2009). We crossed the BcF6-GFP crystal cell marker to a 12xSu(H)-LacZ Notch activation reporter (Go et al, 1998; Duvic et al, 2002) and stained the lymph glands with anti-P1.

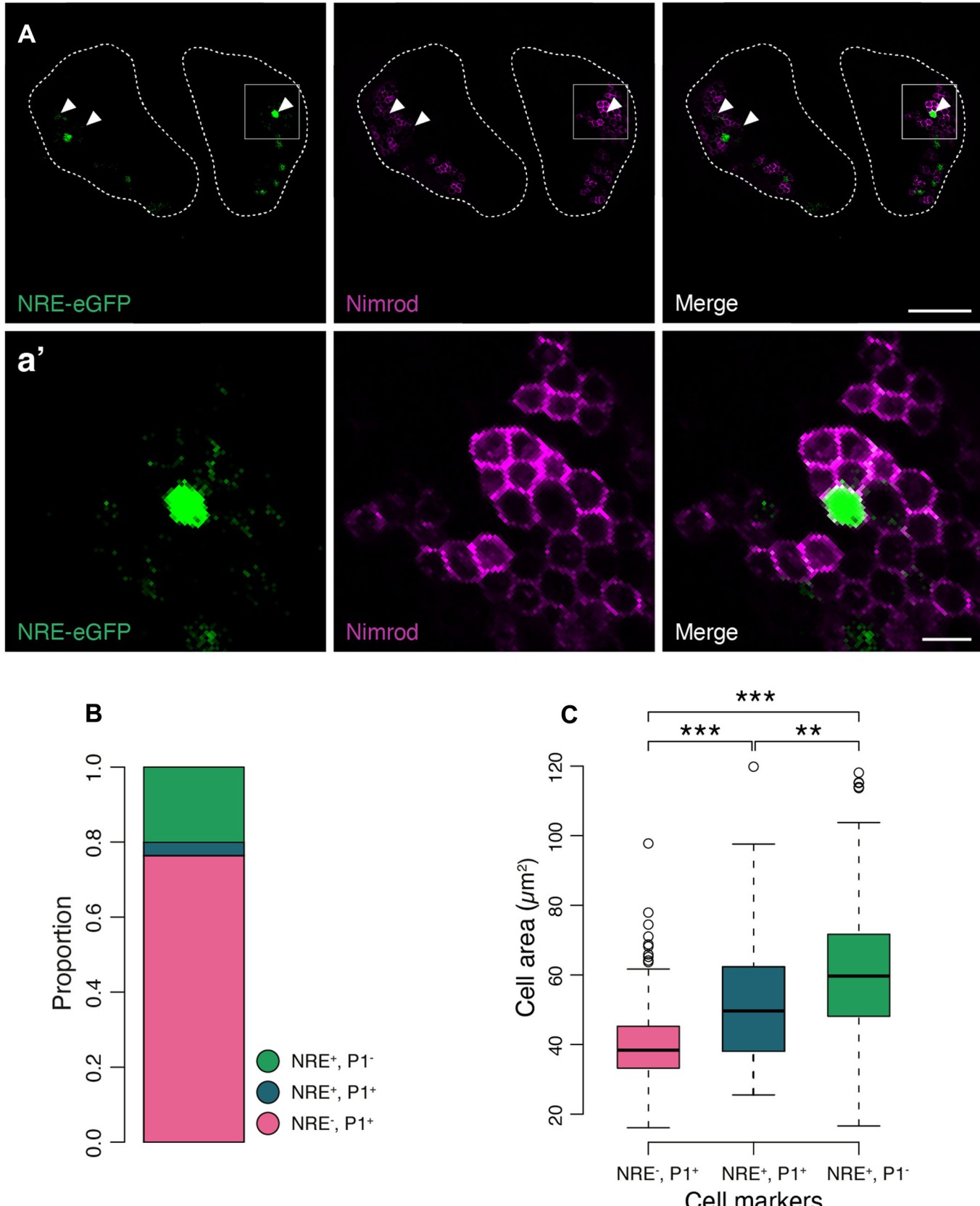

**Figure 1.  Notch signaling mediates transdifferentiation of cortical plasmatocytes into crystal cells.**

(A) Lymph glands from larvae expressing the same NRE-eGFP notch activation reporter were stained with the anti-P1 antibody, a marker of mature plasmatocytes. We observe a population of mature plasmatocytes expressing GFP (white arrowheads), showing that there is a population of Notch-activated plasmatocytes in the cortical region of the lymph glands. Dotted lines define the contour of the primary lymph gland lobes. (a') Tenfold magnification of the inset shown in (A). (B) Quantification of the cells in (B) shows that the majority of marked cells (GFP and P1) are single positive plasmatocytes (76% single positive plasmatocytes), around 3.5% of the marked cells were positive for both markers and the remaining 20% are single GFP positive cells (presumably crystal cells). $n = 761$ cells from four lymph glands. (C) Quantification of the cell area showed a continuum in size from the smallest single P1-positive cells, through to the intermediate double positives and finally the largest single NRE-positive cells. This observation reaffirms the idea that the Notch-activated plasmatocytes are in the process of differentiating into crystal cells. Measurements of cell size were carried out manually by outlining cells in ImageJ at their largest area using Nimrod staining for plasmatocytes and cytoplasmic GFP for Notch activation. Although this is not the ideal, there is no cytoplasmic marker of mature plasmatocytes, and no membrane-localized notch activity reporter. Measuring cell areas using a cytoplasmic GFP will underestimate cell size, which should reduce the effect size observed. (NRE⁻,P1⁺ (582 cells): Min $= 16.08$, Q1 $= 33.22$, Median $= 38.38$, Q3 $= 45.25$, Max $= 61.67$; NRE⁺,P1⁺ (27 cells): Min $= 25.51$, Q1 $= 38.09$, Median $= 49.63$, Q3 $= 63.30$, Max $= 97.59$; NRE⁺,P1⁻ (153 cells): Min $= 16.61$, Q1 $= 48.12$, Median $= 59.66$, Q3 $= 71.69$, Max $= 103.78$). ***$P \leq 0.001$, **$P \leq 0.01$, *$P \leq 0.05$. Cell areas were compared using a one-way ANOVA followed by Tukey's Honest Significant Difference method. The pairwise $P$ values are as follows: NRE⁺,P1⁺ vs NRE⁻,P1⁺ $P = 0.0000003$; NRE⁺,P1⁻ vs NRE⁻,P1⁺ $P = 0.000000$; NRE⁺,P1⁻ vs NRE⁺,P1⁺ $P = 0.011504$. Scale bar represents 100 μm in (A), and 10 μm in (a'). Source data are available online for this figure.

The predicted triple-positive cells could be observed, providing additional validation of transdifferentiating cell presence (Fig. 2B,b').

Again, we could estimate the distribution of the three cell types and show that single positive plasmatocytes constitute circa 80% of the cells, the double-positive transdifferentiating cells are rare (3.3%), and single positive crystal cells account for the rest of the stained population within the cortical zone (Fig. 2C).

Lastly, if crystal cells would derive exclusively from medullary zone precursor cells, the total number of crystal cells should not be affected by Notch signaling blockage in cortical plasmatocytes. In contrast, the transdifferentiation model predicts a reduction of crystal cell numbers if Notch signaling is prevented within these differentiated plasmatocytes. To this aim, we knocked down *Notch* using RNAi driven by *eater*-GAL4, specifically interfering with a necessary crystal cell differentiation process within the plasmatocyte population. We observed a statistically significant reduction of crystal cells when reducing Notch levels relative to a control gene (Wilcox test, $W = 10$, $P = 0.02$) (Fig. 2D).

Taken together, the results above preclude us from rejecting a transdifferentiation process from plasmatocytes into crystal cells within the cortical zone of lymph glands of third-instar larvae.

## Plasmatocyte-to-crystal cell transdifferentiation is a unidirectional process

Although consistent with our transdifferentiation model, the presence of cells transiently expressing both crystal cells and plasmatocytes-specific markers, in the lymph gland, is also compatible with the classic binary fate choice hypothesis. Under this scenario, markers of both cell types are expressed initially following which, one marker is lost, and cell fate is determined. This predicts that double-positive cells would give rise to both single-positive types. In contrast, the transdifferentiation model would predict that double positive cells will only give rise to crystal cells as they constitute an intermediate step in the directional process from plasmatocyte to crystal cell fate.

To test these alternative hypotheses, we performed two symmetrical G-TRACE lineage-tracing experiments, confined to the period from late L2 to late L3, using GAL80ts. The G-trace system permits the tracking of individual cells that have expressed or derive from cells which have expressed a particular driver (GAL4 line) during development. The expression of the driver, in this case,

is limited in time by the presence of a thermo-sensitive GAL80 allele (GAL80ts), that only permits GAL4 activity when the fly is placed at high temperatures. This drives a genomic rearrangement that maintains EGFP expression perpetually under the control of a ubiquitous promoter (Evans et al, 2009). First, we performed lineage-tracing with *eater*-GAL4, a driver expressed in cortical plasmatocytes (Tokusumi et al, 2009) coupled to UAS-FLP, Tub-GAL80ts and UbiFRTstopFRTStinger. In these flies, any cell and its progeny that has expressed *eater* (as mimicked by the *eater*-GAL4 driver line) will display green fluorescence. In addition, we performed antibody staining against Lozenge, a marker of crystal cells, on lymph glands dissected from these individuals. The presence of double-positive cells confirms that *eater*-expressing cells have become crystal cells (Fig. 3A). Complementarily, we did lineage-tracing by crossing an enhancer-trap Lz-Gal4 driver shown to be specific to crystal cells (Terriente-Felix et al, 2013) to UAS-FLP, Tub-GAL80ts, and UbiFRTstopFRTStinger, dissected lymph glands from the progeny and stained them with P1 antibody. In this case, any cell that has expressed Lozenge (crystal cell) will have green fluorescence perpetually and expression of P1 antigen (Nimrod) will indicate plasmatocyte identity. We could not observe double-positive cells in over 340 cells from 9 lymph glands (Fig. 3B). Unlike the previous experiment (Fig. 3A), where a class of double-positive cells could be observed, no cells that have expressed Lozenge (crystal cell fate) show a transition to plasmatocyte fate. Using eater-Gal4 UAS-dGFP (destabilized GFP), we confirmed the very low level of co-localization between a destabilized variant of GFP (dGFP) and Lozenge, revealed with anti-Lz immunostaining (~1.7%, Figs. EV2A,a',a").

G-TRACE lineage-tracing experiments, although highly informative, do not provide a continuous time course for the process in study. To complement our lineage-tracing approach, we optimized a recently established protocol for lymph gland ex vivo culture (Ho et al, 2023), aiming to perform long-term imaging of lymph glands. For this purpose, late L3 larvae (120 h AEL) expressing, simultaneously, markers for plasmatocytes (Eater-dsRed) and crystal cells (BcF6-GFP) were used. In order to conduct the ex vivo assays, late L3 larvae were dissected, and the lymph gland complexes were cultured and imaged for 12 h, at 25 °C. In the end, we were able to image (single and randomly selected) primary lobes of 5 lymph glands, showing the exact moment where single positive Eater-DsRed cells (Eater-DsRed+ BcF6-GFP-) start to express BcF6-GFP, thus becoming double-positive cells (Eater-DsRed+

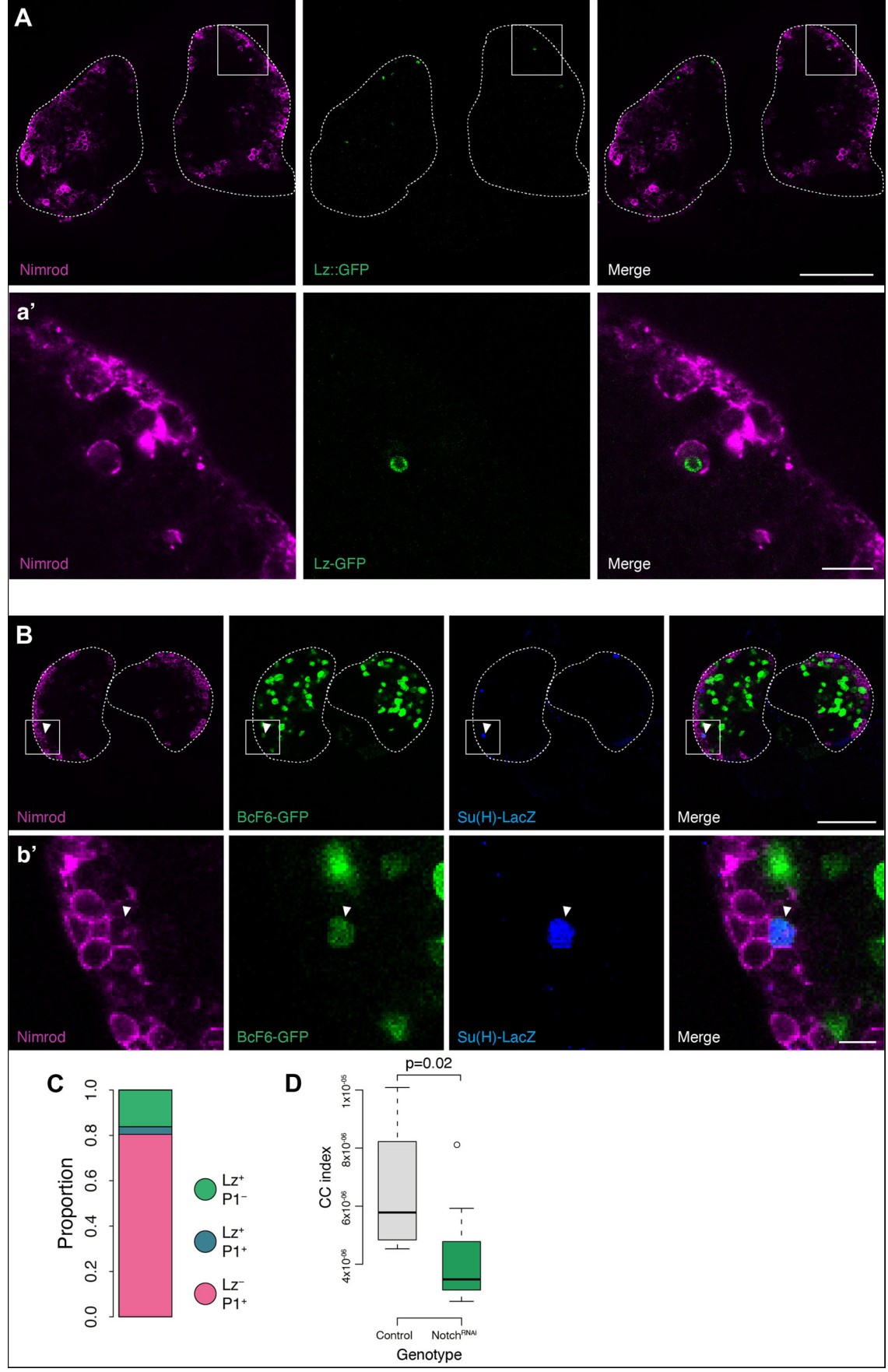

**Figure 2. Notch signaling mediates transdifferentiation of cortical plasmatocytes into crystal cells.**

(A) Lymph glands from larvae expressing a Lz-GFP fusion protein were stained with anti-P1, specific against the Nimrod protein. Cells were observed in the cortical zone expressing both proteins, consistently with the hypothesis that plasmatocytes are transdifferentiating into crystal cells. Dotted lines define the contour of the primary lymph gland lobes. (a′) Tenfold magnification of the inset in (A), showing an example of a double-positive Lz, P1 cell. (B) By combining a BcF6-GFP crystal cell marker with a 12xSu(H)-LacZ Notch activation reporter and staining the lymph glands with anti-P1 and Anti-βgal, we observed triple-positive cells (arrowheads). The existence of this population shows that some plasmatocytes activate the Notch pathway and initiate a crystal cell differentiation program. Dotted lines define the contour of the primary lymph gland lobes. (b′) Tenfold magnification of the inset in (B) showing an example of a BcF6-GFP + , Nimrod+ Su(H)-Lacz+ cell. (C) As in (A), 80% of cells are single positive for P1, 3% are double positive and 16% are single positive for Lz, reaffirming the transient nature of the transdifferentiating double-positive population. (D) Crystal cells differentiate from cortical plasmatocytes through a Notch-dependent mechanism. Crystal cell numbers are reduced when Notch is knocked down in the cortical differentiated plasmatocyte population (eater-GAL4 > UAS-NotchRNAi) compared to an unrelated RNAi control (eater-GAL4 > UAS-CG9313) (Control (8 lymph glands): Min = 4.52e-6, Q1 = 4.84e-06, Median = 5.78e-06, Q3 = 8.23e-06, Max = 1.01e-05; NotchRNAi (8 lymph glands): Min = 2.72e-06, Q1 = 3.11e-06, Median =  3.48e-06, Q3 = 4.78e-06, Max = 5.92e-06). Crystal cell numbers were normalized to lymph gland volume. CC indexes were compared using a Mann–Whitney U test. Scale bars in (A, B) represent 100 µm, and 10 µm in (a′, b′). Source data are available online for this figure.

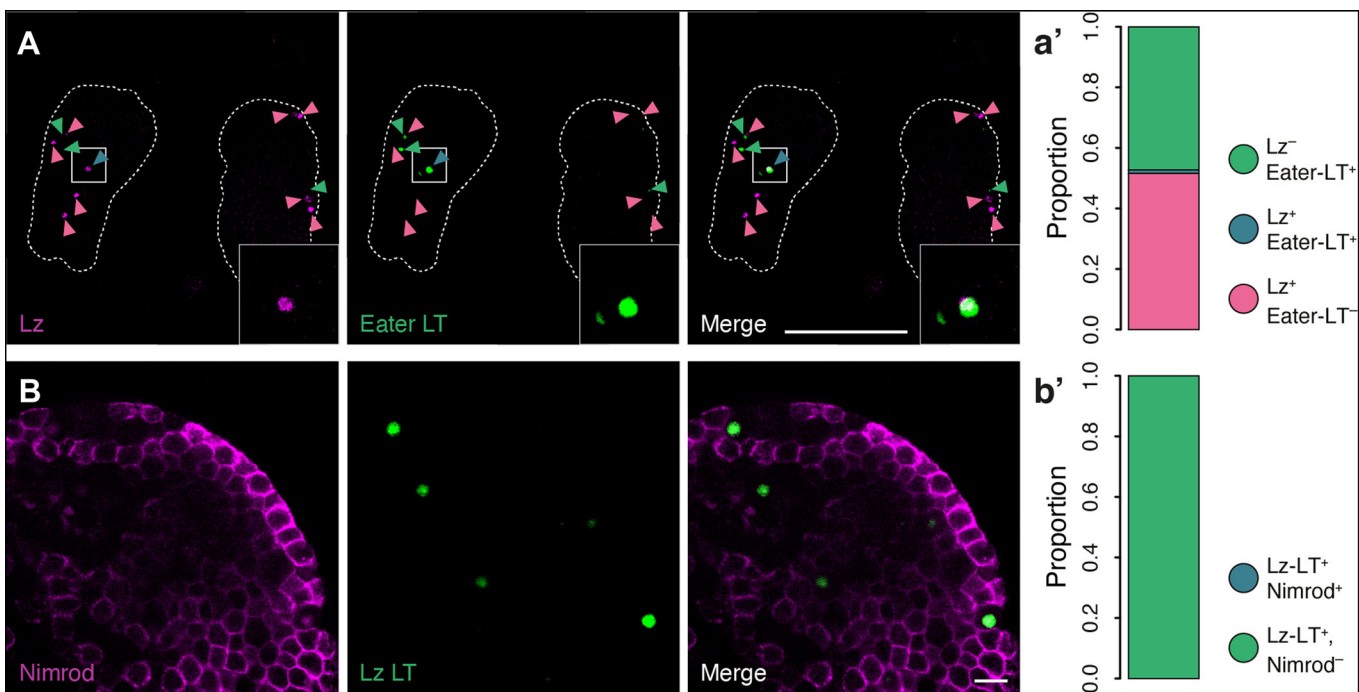

**Figure 3. A unidirectional transdifferentiation program produces crystal cells from plasmatocytes.**

(A) Lineage tracing of Eater-Gal4 followed by staining with anti-Lz. Double-positive cells are observed, indicating that some crystal cells originated from Eater-positive precursors. Magenta arrowheads mark Lz+ nuclei, green arrowhead mark cells lineage-traced for Eater-Gal4 but are Lz-, and cells indicated by blue arrowheads indicate cells positive for both (inside inset). Dotted lines define the contour of the primary lymph gland lobes. (a′) Quantification of the three cells types shown in (A). (B) By driving lineage tracing for Lz during larval development (limited with Tub-Gal80ts) and staining the lymph glands with anti-P1, lineage-traced cells were invariably negative for Nimrod (P1) (quantified in (b′)). This supports the idea that the double-positive cells do not revert back to plasmatocyte fate, and instead become crystal cells. (b′) Quantification of the three cells types shown in (B). Scale bars in (A) represent 100 µm and 10 µm in (B). Source data are available online for this figure.

BcF6-GFP +) (see Movie EV1). In accordance with the G-TRACE lineage-tracing experiments, no mature crystal cells (Eater-DsRed-BcF6-GFP +) were ever observed to become double positive. As such, we reinforce that these double-positive population likely represent cells undergoing unidirectional, plasmatocyte-to-crystal cell transdifferentiation. Due to technical difficulties, only four out of the five imaged lobes were quantifiable, for the proportion of cells within each of the three cell populations. Nonetheless, out of 1264 imaged cells, from 4 single lobes, belonging to 4 independent lymph glands: circa 96% of the cells were plasmatocytes, about 2.5% were crystal cells and only 1.5% were observed to become double-

positive, transdifferentiating cells—further attesting their rareness. Interestingly, at a point in time, some of these imaged double-positive cells momentarily acquired a "dark spot" within their BcF6-GFP expression domain. We speculate that these "spots" may be PPO crystals in production, which would reflect the functional switch between the two cell types (Fig. EV3).

Morphological analysis of these samples with segmented and tracked nuclei allowed us to determine where new crystal cells are differentiating in the lymph gland (Fig. EV4). We could confirm the presence of transdifferentiating cells within the outer 2 cell layers, a region that corresponds to the cortical zone. This observation is

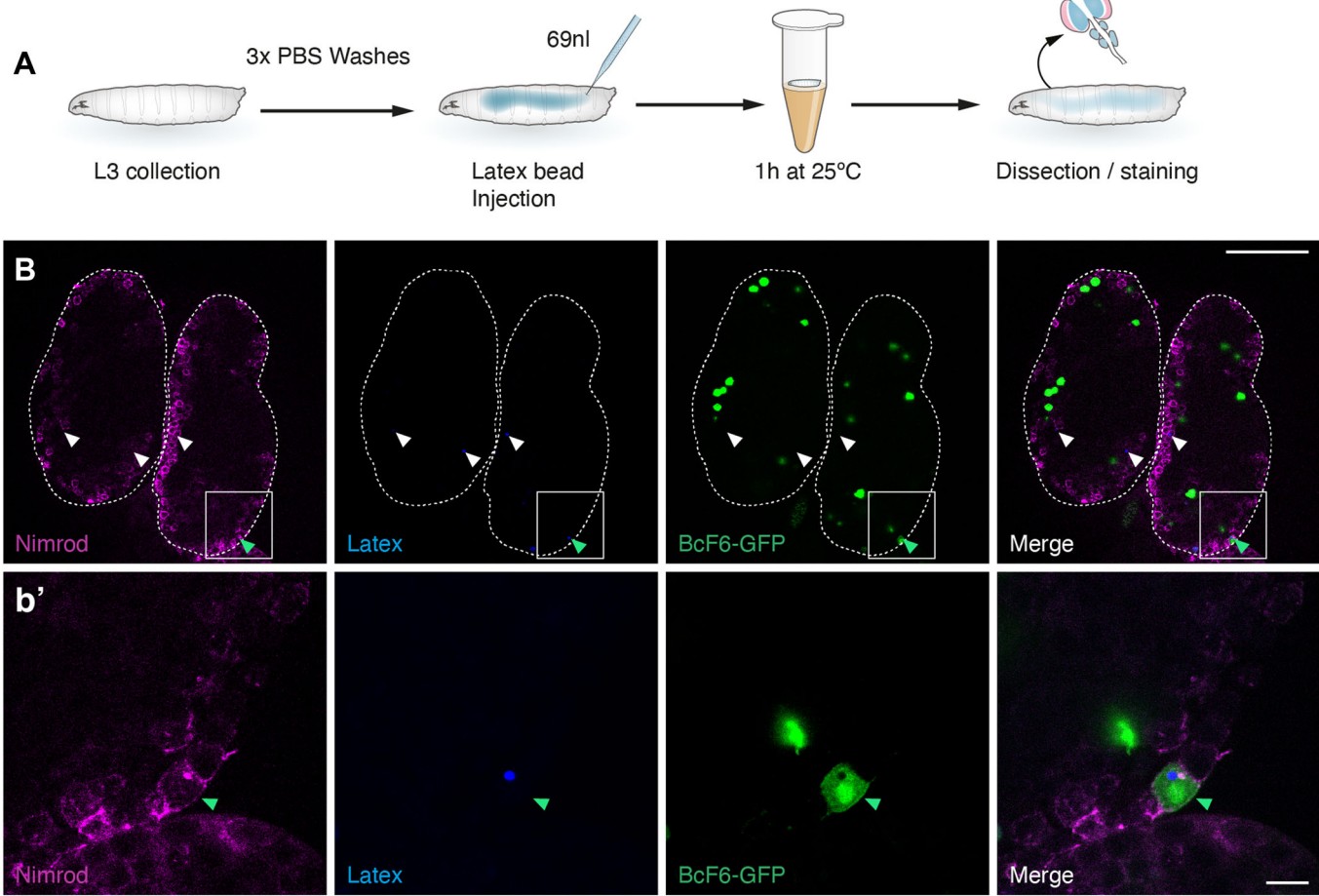

**Figure 4. Cortical plasmatocytes are phagocytic cells that can transdifferentiate into crystal cells.**

(A) Brief protocol of the in vivo phagocytosis assay. (B) We can detect fluorescent beads inside double-positive cells for P1 (plasmatocyte) and BcF6-GFP (crystal cell) (white and green arrowheads), confirming that active phagocytes can engage into a transdifferentiation process towards crystal cell fate. Dotted lines define the contour of the primary lymph gland lobes. (b') A zoomed region (inset) of (B) pointing with a green arrowhead a cell expressing Nimrod, BcF6-GFP and containing a phagocytosed latex bead (blue channel). Scale bar represents 100 μm in (B) and 10 μm in (b'). Source data are available online for this figure.

crucial because eater-dsRed, which uses the same enhancer as eater-GAL4, is expressed in distal progenitors of the intermediary zone (Blanco-Obregon et al, 2020). Consequently, our observations from the *Notch* knockdown experiment (Fig. 2D) and the analysis of *eater* lineage tracing (Fig. 3) could be explained by events occurring solely the distal part of the intermediary zone.

All of the above observations are consistent with a process in which mature plasmatocytes (P1-positive cells) activate the Notch pathway to initiate a crystal cell differentiation program consisting in the initiation of Lozenge and PPO1 expression, whilst shutting down plasmatocyte-specific genes such as Eater and Nimrod.

## Transdifferentiating cells were once phagocytically active

We have shown that established bona fide markers of both plasmatocyte and crystal cell fate co-exist in the same cell population and confirmed the unidirectionality of the process. To complement these findings, we proceeded to test the functional transition from phagocytosis, the hallmark of fully differentiated

plasmatocytes, to a phenoloxidase-producing role. Our model predicts that some of these phagocytic cells will become crystal cells, transitioning from a Nimrod-positive to a Bc-positive state. To test this prediction, we performed in vivo and ex vivo phagocytic assays.

For the in vivo assay, blue fluorescent beads were injected into BcF6-GFP third-instar larvae (Fig. 4A)—similarly to an established protocol (Horn et al, 2014; preprint: Corcoran et al, 2020). The gene Bc (black cells) or PPO1 codes for prophenoloxidase 1, and is a marker of larval crystal cells (Rizki and Rizki, 1980a; Neyen et al, 2015). Dissected lymph glands at 1-h post-injection were stained with P1 antibody, allowing detection of both single positive populations and of the predicted double-positive population of transdifferentiating cells (Fig. 4B,b'). We observed beads inside plasmatocytes exclusively within the cortical region, as expected. More importantly, we could confirm the existence of double-positive cells containing beads (Fig. 4B,b'), providing qualitative evidence that a phagocytic cell, which has expressed plasmatocyte differentiation markers becomes a crystal cell. As a control and to assess the specificity of the phagocytic capacity, we mounted eye

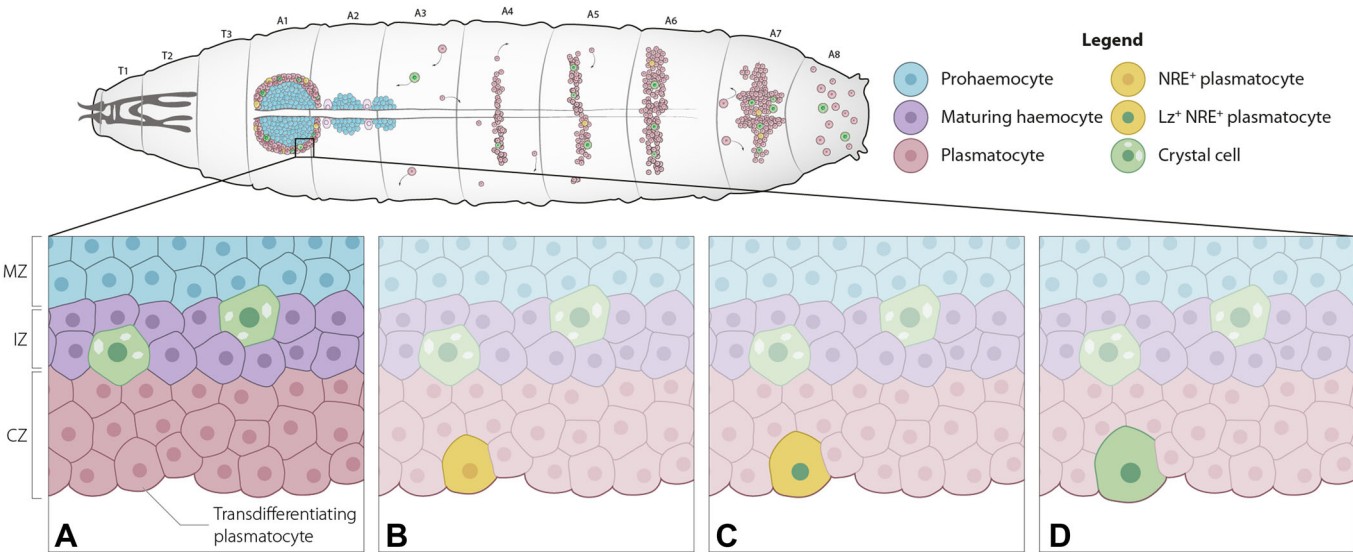

**Figure 5. A model for crystal cell differentiation in the larval lymph gland.**

Cortical plasmatocytes (red cells) derived from medullary prohaemocytes (blue cells) can engage in a Notch-dependent transdifferentiation process. A temporal notion of this process is provided in the succession of panels from (**A**) to (**D**), whereby Notch-activated plasmatocytes (yellow cells (**B**)) transiently express both cell type markers (yellow cell with green nucleus (**C**)), and become crystal cells (green cells (**D**)). These same cells and processes are shown in the clusters alluding to previously published data (Leitão and Sucena, 2015). Source data are available online for this figure.

discs which are surrounded by a patch of haemocytes (Holz et al, 2003). We can confirm that only haemocytes display internalized beads by comparison to the eye disc (ED) and optic lobe (OL) that do not contain beads (Fig. EV5).

As before, the presence of a double-positive population constitutes a qualitative criterion that precludes the rejection of the transdifferentiation model. Therefore, our observations corroborate further a mechanism through which crystal cells producing phenoloxidase may stem from the transdifferentiation of phagocytic Nimrod-positive cells.

## Discussion

The overwhelmingly prevalent mode of crystal cell differentiation in the lymph gland is through a binary choice that occurs in immature haemocytes (progenitors), in a *Notch*-dependent manner (reviewed in Banerjee et al, 2019). However, in light of the evidence gathered here, we propose an additional mechanism for the production of crystal cells in the lymph gland of homeostatic larvae (Fig. 5). We posit that, as in the sessile clusters (Leitão and Sucena, 2015) and dorsal vessel (Cevik et al, 2019), crystal cells can be generated through the transdifferentiation of another functionally mature cell type, the plasmatocyte. This model is supported further by recent transcriptomics studies, showing crystal cell differentiation from haemocyte clusters expressing mature plasmatocyte markers, such as Eater and NimC1 (Cho et al, 2020; Girard et al, 2021; Hultmark and Andó 2022). During crystal cell differentiation, these markers are downregulated while mature crystal cell markers are upregulated (Cho et al, 2020; Girard et al, 2021). This mechanism parallels previously described transdifferentiation of plasmatocytes into crystal cells in the sessile haemocyte

clusters (Leitão and Sucena, 2015; preprint: Corcoran et al, 2020) and extends the established plasmatocyte transdifferentiation into lamellocytes within and outside the lymph gland (Honti et al, 2010; Anderl et al, 2016; Cho et al, 2020; Tattikota et al, 2020). Our observations—allied to these findings—support the occurrence of plasmatocyte-to-crystal cell transdifferentiation in the lymph gland, at multiple levels. Morphologically, we showed a transition in cell size throughout this process that matches the classical description of these cell types; functionally, lineage-tracing revealed the past phagocytic capacity of differentiating crystal cells, and through ex vivo long-term live imaging, the shift from plasmatocyte fate (shut-off) to crystal cell markers (activation). We argue that this transition from the expression of terminal markers of one cell type into another, concurrent with radical morphological and functional changes, support a transdifferentiation process rather than a classification of such cells as oligopotent.

Other important parallels have been established previously, namely the dependence on Notch/Serrate signaling for crystal cell differentiation in diverse contexts (Duvic et al, 2002; Lebestky et al, 2003; Leitão and Sucena, 2015; Cevik et al, 2019; Blanco-Obregon et al, 2020). Using a Serrate protein trap line (Nagarkar-Jaiswal et al, 2015), our observation of Serrate expression predominantly in the most external part of the cortical region, suggests this mechanism operates in the outer layer of the lymph gland (Fig. EV6). This outer layer is composed of mature cells (Jung et al, 2005; Crozatier and Meister, 2007), mostly plasmatocytes. Our data are consistent with this idea as shown by confined expression of markers for both plasmatocytes and crystal cells as well as for Notch pathway activation (Figs. 1, 2, and EV1). In brief, we argue that, for both the sessile clusters and the lymph gland, the Notch/Serrate pathway acts in interacting plasmatocytes to promote their transdifferentiation into crystal cells.

With the establishment of live imaging methods in the lymph gland, future studies may focus on the quantification of the duration of the transition state (double-positive cells). Despite not being able to observe the moment in which Eater-dsRed expression is "shut-down" in our experiments, we argue that Eater-dsRed might have been slow to denaturate in our setup, giving the impression of a longer duration of the transition state. Given the low proportion of the double-positive cell type in the overall haemocyte population (Figs. 1B, 2C, and 4), we predict that the transition state is likely to be a very fast process.

We interpret the higher proportion of crystal cells than expected (circa 20%) as a result of an under-estimation of plasmatocyte number, given that P1 is a late plasmatocyte marker and therefore provides a conservative estimate of plasmatocyte numbers. Indeed, plasmatocytes could be differentiated and active before strong and consistent expression of Nimrod and transdifferentiate before the detection of P1.

The parallels drawn here, between both haematopoietic contexts, may permit blood cell fate decisions to be addressed in the tractable sessile clusters and later validated in the lymph gland. Indeed, the generalization of a transdifferentiation mechanism across hematopoietic larval organs/tissues requires additional mechanistic evidence. Coupling the advances in in vivo imaging with the recent scRNA-seq studies of the lymph gland that have significantly advanced our understanding of blood cell differentiation (Cho et al, 2020; Tattikota et al, 2020; Girard et al, 2021), points the way to further studies. However, it is worth noticing that a considerable disparity in resolution remains between the subpopulations identified in scRNA-seq data and those marked by GAL4 drivers in the lymph gland. Hence, to validate the developmental pathways proposed in these studies, it is essential to develop new drivers that specifically target distinct blood cell subpopulations, which may require combinatorial strategies, such as the split-GAL4 system (Luan et al, 2006). This strategy may be applied to questions about the dynamics of cell differentiation over time and space, where live imaging is essential. For example, in the clusters the rate of import/export defines their size and the concomitant number of crystal cells, a seemingly central mechanism to the regulation of blood cell proportions (Leitão and Sucena, 2015). We speculate on an equivalence between the rate of crystal cell differentiation in the lymph gland as a function of the rate of plasmatocyte export to the cortical zone, and the rate of adhesion of circulating haemocytes to the sub-epidermal sessile clusters.

Notwithstanding the prevalence of binary switch mechanisms in cell fate decision (Cho et al, 2020), including in the lymph gland (Lebestky et al, 2000; Duvic et al, 2002; Banerjee et al, 2019), our findings may constitute an additional layer to this well-established canonical differentiation process. Importantly, these findings add to the transdifferentiation from plasmatocytes to crystal cells described in sessile clusters and the dorsal vessel (Leitão and Sucena, 2015; Cevik et al, 2019; preprint: Corcoran et al, 2020) supporting the generality of this mechanism in Drosophila larvae. Regardless of the size of the contribution transdifferentiation may have in normal lymph gland development and homeostatic conditions, the importance it may play if such conditions are under challenge, remains to be determined. Interestingly, two recent studies have shown that oxygen sensing may be at play, in promoting crystal cell differentiation in the lymph gland

(Mukherjee et al, 2011; Cho et al, 2018; Shin et al, 2024). As the role of oxygen sensing in promoting plasmatocyte-to-crystal cell transdifferentiation has already been shown in the sessile clusters surrounding the caudal sensory cones (preprint: Corcoran et al, 2020), it is possible that the same could happen in the context of the lymph gland. In addition, in the absence of PPO2, larvae display a physiological state of hypoxia even under normoxic conditions as this product of crystal cells fails to ensure its role in haemocyte guidance and systemic oxygen circulation (Shin et al, 2024). In this light, it will be interesting to re-visit the relative weight of transdifferentiation under hypoxic conditions, and whether this mechanism may be called upon with higher prevalence under stress. Future inquiries will determine whether transdifferentiation of plasmatocytes into crystal cells is a general mechanism, under local or systemic stress, to increase the deployment and response capacity of these cells (Binggeli et al, 2014; Dudzic et al, 2015; Vlisidou and Wood, 2015; Letourneau et al, 2016; Cho et al, 2018; Ramesh et al, 2021). Such future enquiry holds the promise of extending fundamental knowledge, gathered over more than half a century, on crystal cell differentiation and function in response to external aggression to promote melanization and wound-healing (Rizki, 1959; Rizki, and Rizki, 1980b; De Gregorio et al, 2002; Sorrentino, et al, 2002; Bidla et al, 2007; Eleftherianos and Revenis 2010; Dudzic et al, 2015; Banerjee et al, 2019; Shin et al, 2024).

As haematopoiesis in the Drosophila adult remains contentious, even after infection (Ghosh et al, 2015; Sanchez Bosch et al, 2019; Boulet et al, 2021), another aspect unaddressed is whether transdifferentiation occurs at this stage. The existence of such a mechanism could ensure plasticity in the regulation of haemocyte-type availability in the adult. Finally, a question remains on how these two processes may interplay and be modulated under different immunological and/or metabolic conditions (Ramond et al, 2020; Tiwari et al, 2020).

Transdifferentiation is gaining momentum, particularly in the context of development, regeneration, disease, and homeostatic balance (Slack, 2007; Thorel et al, 2010; Richard et al, 2011; Goldman, 2014; Defourny et al, 2015; Merrell and Stanger, 2016; Schaub et al, 2018; Wei et al, 2020). Albeit most reported cases involve cell division, a few exceptions have reported direct transdifferentiation, as described in our system (Beresford, 1990; Thorel et al, 2010; Jopling et al, 2011; Wei et al, 2020). Indeed, the reported switch from plasmatocyte to crystal cell fate in the lymph gland and in the sessile clusters obeys the three criteria postulated for this mechanism (Beresford, 1990): (i) cells are functionally mature, (ii) they change phenotype and, (iii) do not divide before transdifferentiating. In highly spatially structured systems such as the haematopoietic system of both Drosophila and vertebrates, particularly in its post-natal dimension, transdifferentiation may play a more prevalent role than previously appreciated. Finally, the generality of the transdifferentiation mechanisms in this system is of potential relevance to further develop genetic models of the vertebrate hematopoietic lineages (Crozatier and Vincent, 2011; Gold and Brückner, 2015; Banerjee et al, 2019), known to be highly context-dependent and plastic (Xie et al, 2004; Zhou et al, 2009; Sica and Mantovani, 2012; Hori, 2014; Dzierzak and Bigas, 2018) as well as for the study of regeneration and pathology (Slack, 2007; Jopling et al, 2011; Goldman, 2014; Zhou et al, 2014; Tanabe et al, 2018; Mollinari et al, 2018; Wei et al, 2020).

# Methods

### Reagents and tools table

| Reagent/resource | Reference or source | Identifier or catalog number |
|---|---|---|
| **Experimental models** | | |
| Drosophila Outbred population | Faria and Sucena, 2017 | |
| HmlΔ-cytoDsRed | Clark et al, 2011 | |
| BcF6-GFP | Gajewski et al, 2007 | |
| Eater-Gal4 | Tokusumi et al, 2009 | |
| w; Eater-dsRed; BcF6-GFP | | |
| 12xSu(H)-LacZ | | |
| HmlΔ-Gal4 | Bloomington Drosophila Stock Center | BL-6395 and BL-6396 |
| Lz-Gal4,UAS-GFP | Bloomington Drosophila Stock Center | BL-6313 |
| NRE-eGFP | Bloomington Drosophila Stock Center | BL-30727 |
| Lz-GFP.FPTB/TM6b | Bloomington Drosophila Stock Center | BL-43954 |
| Ser-eGFP | Bloomington Drosophila Stock Center | BL-42314 |
| w; Tub-Gal80ts; UAS-FLP, UbiFRTstopFRTStinger | Bloomington Drosophila Stock Center | BL-28282 |
| UAS-NotchRNAi | Vienna Drosophila Stock Center | KK100002 |
| UAS-SerRNAi | Vienna Drosophila Stock Center | KK108348 |
| UAS-CG9313RNAi | Vienna Drosophila Stock Center | KK 103600 |
| **Antibodies** | | |
| Anti-Lozenge (Mouse) | DSHB | AB528346 |
| Anti-NimrodC1 (Mouse) | Istvan Andó, Institute of Genetics, Szeged, Hungary | P1 antibody |
| Anti-GFP (Rabbit) | Torrey Pines Biolabs | TP401 |
| Anti-βgal (Rabbit) | ThermoFisher | A-11132. |
| Anti-Mouse Alexa fluor 488 | Invitrogen | A-11001 |
| Anti-Mouse Alexa fluor 546 | Invitrogen | A-11008 |
| Anti-Rabbit Alexa fluor 488 | Invitrogen | A-11001 |
| Anti-Rabbit Alexa fluor Cy5 | Invitrogen | A-10523 |
| **Other** | | |
| Amine-modifiedpolystyrene, fluorescent blue Latex beads | Sigma-Aldrich | L0280 |

## Stock maintenance

Flies were kept under room temperature and fed with a standard cornmeal-agar medium, consisting of 4.5% molasses, 7.5% sugar, 7% corn-flower, 2% granulated yeast extract, 1% agar and 0.25% nipagin, mixed in distilled water. Experiments were carried out at 25 °C, apart from the lineage tracing in which egg lays were kept at 18 °C during the restrictive phase and moved to the permissive 29 °C at 44 h after egg lay. For RNAi experiments, egg lays were carried out at 25 °C following which the eggs were moved to 29 °C. For experiments with the Lz-GFP.FPTB/TM6b, we outcrossed the stock and selected non-tubby larvae for observation.

## Fly stocks

The following fly stocks were used: Drosophila Outbred population (Faria and Sucena, 2017), HmlΔ-cytoDsRed (Clark et al, 2011) (a gift from Marc Dionne), BcF6-GFP (Gajewski et al, 2007) and Eater-Gal4 (Tokusumi et al, 2009), w; Eater-dsRed; BcF6-GFP (a gift from Robert A. Shultz); from the Bloomington stock center: HmlΔ-Gal4 (BL-6395 and BL-6396), Lz-Gal4 was derived from Lz-Gal4, UAS-GFP (BL-6313) (Crew et al, 1997), NRE-eGFP (Saj et al, 2010) (BL-30727), 12xSu(H)-LacZ (Go et al, 1998), Lz-GFP.FPTB/TM6b (BL-43954), Ser-eGFP (Nagarkar-Jaiswal et al, 2015) (BL-42314); from the VDRC stock center: UAS-NotchRNAi (KK100002), UAS-SerRNAi (KK108348) and UAS-CG9313RNAi (KK 103600). To carry out temperature-dependent lineage tracing, females of w; Tub-Gal80ts; UAS-FLP, UbiFRTstopFRTStinger (from Bloomington stock BL-28282) were crossed with males from the Lz-Gal4 driver at 18 °C. Only females were analyzed.

## Immunostaining

Lymph gland dissections and immunostaining were carried out following the protocols described by (Evans et al, 2014). Briefly, for all experiments, wandering stage L3 larvae (120 h AEL) were washed 3× in 1×PBS to remove yeast from the cuticle. Larvae were dissected in cold PBS and the lymph glands transferred with the mouth hooks and brain complex to 1 ml of 0.002% ice-cold PBST× (Evans et al, 2014). After 30 min of dissection, the lymph glands were fixed in 3.7% formaldehyde. After 3 × 15-min washes in 0.4× PBST×, the lymph glands were blocked in 5x blocking buffer (5×NDS in 0.4×PBST×), and subsequently incubated in primary antibodies diluted in blocking buffer. Both the primary and secondary antibody incubations were carried out overnight at 4 °C.

The following primary antibodies were used in the following concentrations:

1:100 Anti-P1 (Mouse), 1:500 Anti-GFP (Rabbit), 1:200 Anti-βgal (Rabbit), 1:100 Anti-Lz (Mouse).

The following secondary antibodies were used in the indicated dilutions:

Anti-Mouse af488 (1:250), Anti-Mouse af546 (1:250), Anti-Rabbit af488 (1:500), Anti-Rabbit afCy5 (1:250).

Samples were mounted in Vectashield then imaged using a Leica SP5 confocal using either a ×40 dry, ×40 oil immersion or ×20 dry objectives. Image processing and quantification were carried out in Fiji (Schindelin et al, 2012) with the Bio-Formats plugin (Linkert et al, 2010).

## Ex vivo lymph gland long-term imaging

Lymph gland dissections, culturing and imaging were carried out following the protocols described by (Ho et al, 2023). Briefly, fate third-instar larvae (200 AEL), from a *w; Eater-dsRed; BcF6-GFP* line, were selected and washed, then dissected in in Drosophila Schneider's medium at room temperature. The connection between target organs— including larval lymph gland, fat bodies, central nervous system and

heart tube - was maintained during all steps—from dissection, to mounting and imaging—for structural integrity purposes. The dissected organ complexes were mounted in a glass bottom dish with PBS, over a 0.1% L-polylysine coat, and cultured in Schneider's medium supplied with 15% FBS (ThermoFisher Scientific, #12483-020) and 0.2 mg/mL insulin (Sigma I0516); the culture medium was prepared fresh, 10 min prior to dissection. The organ-complex was then covered with a 1% agar pad, and 2 ml of the medium, in order to maintain optimal moisture conditions during live imaging. All live imaging experiments were performed for 12 h, at 25 °C, using an LSM980+Airyscan inverted light sheet microscope, with a ×40 water immersion lens. The obtained lymph gland z-sections were spaced by 1 μm, and imaged at a 5- (first movie obtained) or 10-minute (remaining 4 movies obtained) intervals, using excitation at 488 nm (green laser) and 561 nm (red laser). Time-lapse recordings of lymph glands, and resulting t-series images, were processed using Fiji (Schindelin et al, 2012) and IMARIS (Oxford Instruments) to segment and track nuclei over time.

### In vivo phagocytic assays

L3 larvae from a BcF6-GFP line introgressed into an outbred background were collected and washed 3× in 1×PBS. 2 μm of blue fluorescent latex beads (Sigma-Aldrich) were diluted 1:10 in 1×PBS and 69 nl of the suspension was injected into the posterior of the larva at 230 nl/second using a microinjector (Drummond Nanoject II Auto-Nanoliter Injector). The larvae were transferred to food and incubated for 1 h at 25 °C, after which they were collected, their lymph glands dissected and stained with Anti-P1 (as above).

### Statistical analysis

Statistics and graphics were produced in R (version 3.3.0) (R Core Team, 2016). ANOVA followed by Tukey's HSD post-hoc test was used to compare the cell areas of the NRE and Nimrod-positive populations. Wilcox test was used to compare the crystal cell index in the Notch knockdown experiment.

## Data availability

Source data is available at: https://drive.google.com/drive/folders/1pKtjfpmczLmOZSxcf2kKnKum-0-XGqPq?usp=sharing.

The source data of this paper are collected in the following database record: biostudies:S-SCDT-10_1038-S44319-025-00366-z.

## Peer review information

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

## Acknowledgements

The authors are indebted to Cory Evans for sharing the unpublished protocol used in the in vivo phagocytosis experiment. The authors thank Liliana Vieira, Ana Catarina Morais, and Joana Carvalho for technical assistance and input. The authors warmly thank for their generosity, István Andó for sharing the P1 antibody, and Jiwon Shim for putting together and making publicly available the Fly scRNA-seq Database. The Lozenge antibody developed by Utpal Banerjee was obtained from the Developmental Studies Hybridoma Bank, created by the NICHD of the NIH and maintained at The University of Iowa, Department of Biology, Iowa City, IA 52242. For fly stocks, we thank Bruno Lemaitre, Utpal Banerjee, Marc Dionne and Robert A. Schulz, as well as the Bloomington Drosophila Stock Center (NIH P40OD018537). This work was supported by Instituto Gulbenkian de Ciência/Fundação Calouste Gulbenkian and FCT-Fundação para a Ciência e a Tecnologia (Portugal) UIDB/00329/2020 and by C.S. CONGENTO, project LISBOA-01-0145-FEDER-022170, co-financed by Lisboa Regional Operational Programme (Lisboa 2020), under the Portugal 2020 Partnership Agreement, through the European Regional Development Fund (ERDF), and Foundation for Science and Technology (Portugal).

## Author contributions

**Julien Marcetteau**: Conceptualization; Formal analysis; Investigation; Visualization; Methodology; Writing—original draft; Writing—review and editing. **Patrícia Duarte**: Conceptualization; Formal analysis; Investigation; Visualization; Methodology; Writing—original draft; Writing—review and editing. **Alexandre B Leitão**: Conceptualization; Formal analysis; Supervision; Methodology; Writing—original draft; Writing—review and editing. **Élio Sucena**: Conceptualization; Supervision; Funding acquisition; Methodology; Writing—original draft; Project administration; Writing—review and editing.

Source data underlying figure panels in this paper may have individual authorship assigned. Where available, figure panel/source data authorship is listed in the following database record: biostudies:S-SCDT-10_1038-S44319-025-00366-z.

## Disclosure and competing interests statement

The authors declare no competing interests.

# Expanded View Figures

**Figure EV1.  Notch activity in cortical plasmatocytes and co-localization of P1 with crystal cell markers.**

(**A**) Hml-CytoplasmicDsRed was used to mark mature haemocytes of the cortical region and NRE-eGFP as a reporter of Notch activation. NRE-eGFP expression is restricted to the cortical region of the lymph gland showing that Notch activation is restricted to mature and maturing plasmatocytes. Beneath the main image is a XZ slice of the same lymph gland illustrating the specificity of the two markers to the cortical region. Dotted lines define the contour of the primary lymph gland lobes. (**a'**) Shows an orthogonal section of the Lymph glands shown in (**A**). The horizontal dashed line in (**A**) represents the location of the slice. (**B**) Bc-GFP lymph glands stained with anti-Nimrod (red). Bc-GFP is a marker of crystal cells that co-localizes in a subset of cortical cells with the plasmatocyte marker Nimrod (P1) (white arrowheads). Dotted lines define the contour of the primary lymph gland lobes. (**b'**) 10-fold magnification of the inset in (**B**). (**C**) Lz-Gal4, UAS-GFP lymph glands stained with P1 (anti-Nimrod) (red). In agreement with the staining in (**A**), double-positive cells can be observed in the cortical region of the lymph gland (white arrowheads). Dotted lines define the contour of the primary lymph gland lobes. (**c'**) 10-fold magnification of the inset in (**C**). Scale bars represent 100 μm in (**A–C**), and 10 μm in (**a'–c'**).

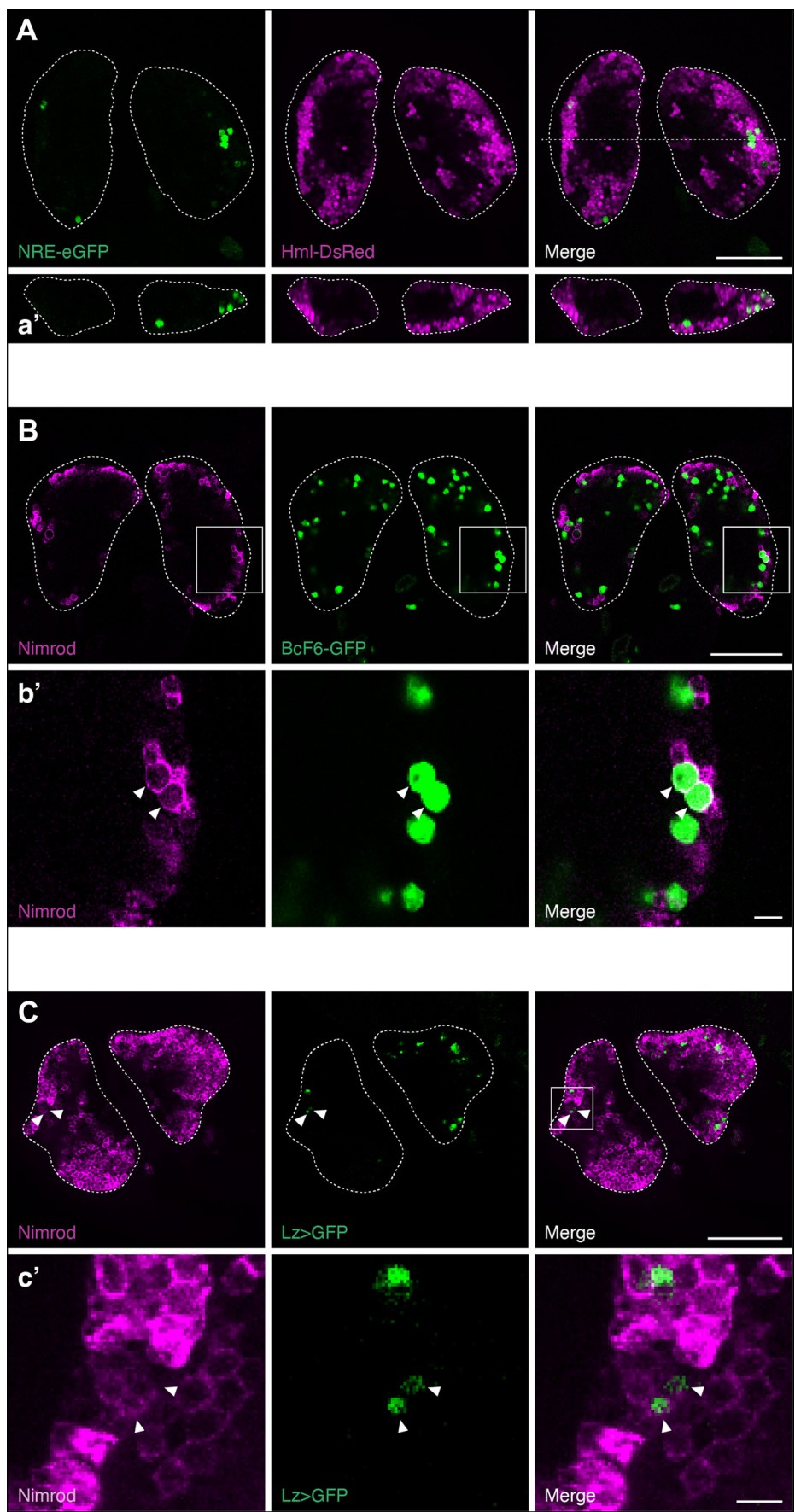

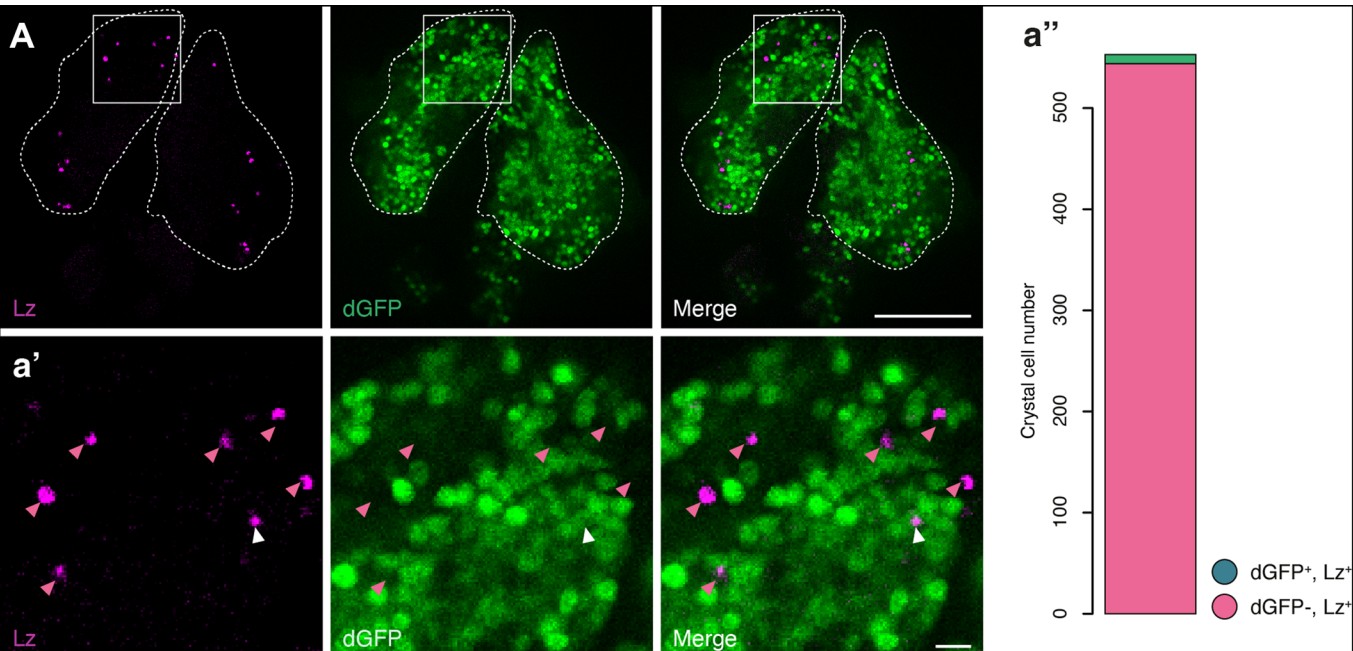

**Figure EV2.  Eater-Gal4 expression is rapidly lost in Crystal cells.**

Eater-Gal4 driving destabilized GFP expression is lost rapidly in crystal cells as shown by the lack of co-localization with Lozenge. Dotted lines define the contour of the primary lymph gland lobes. (a') 10-fold magnification of the inset in (A). Magenta arrows mark cells positive for Lz alone. Rare double-positive cells are marked by white arrows. (a'') Scale bar represents 100 μm in (A) and 10 μm in (a').

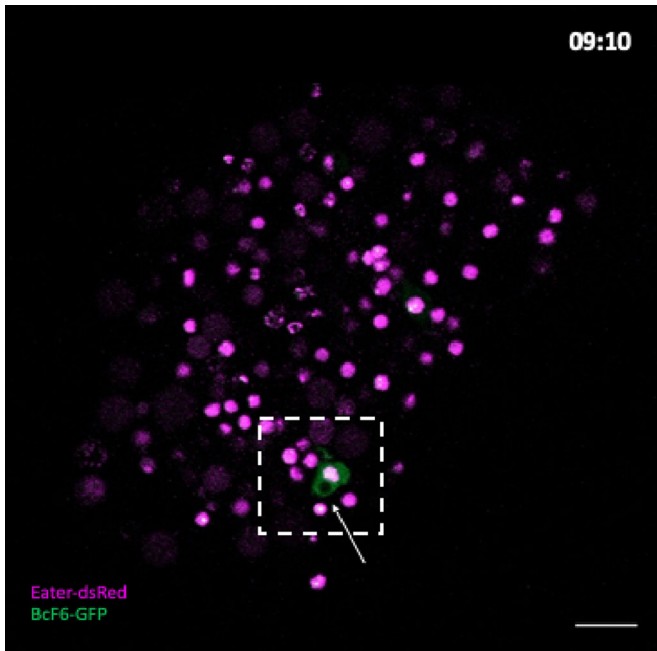

**Figure EV3.  "Dark spots" observed in double-positive cells from cultured lymph glands.**

Image extracted from timelapse represented in Movie EV1. At around 9 h of culture, "dark spots" were observed within the BcF6-GFP expression domain of a double-positive cell (inside white dotted box), suggesting PPO1 production by transdifferentiating cells. Scale bar represents 20 μm.

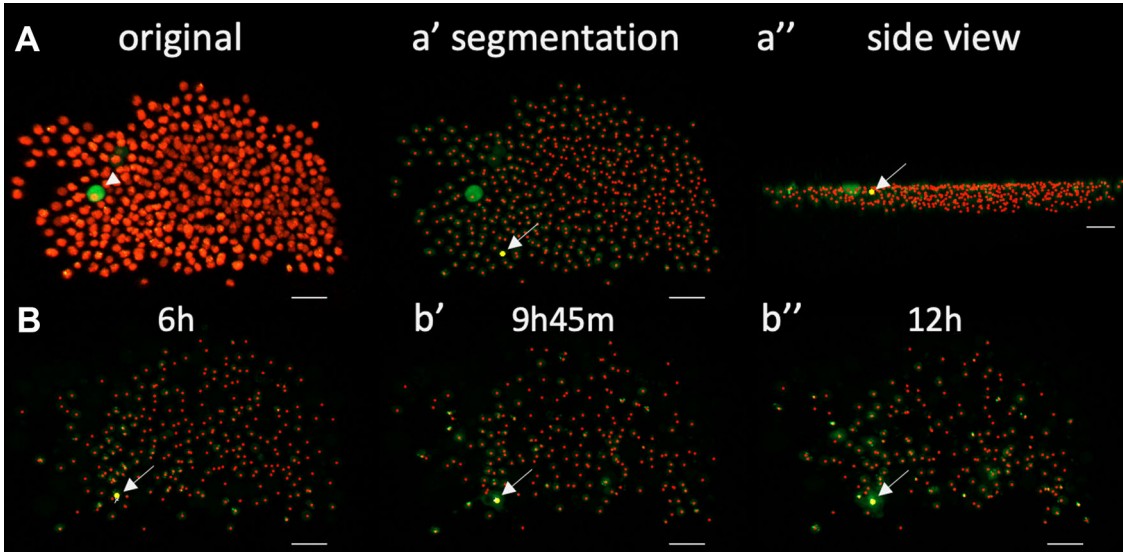

**Figure EV4.  Transdifferentiation of plasmatocytes into crystal cells can occur in the cortical zone of the lymph gland.**

(**A**) Lymph gland from Movie EV1 with *Eater*-DsRed and BcF6-GFP markers. Crystal cells can be observed with cytoplasmatic GFP signal (green arrowhead). (**a'**) Spot segmentation of DsRed nuclei allows the tracking of individual cells throughout the experiment. Two *Eater*-DsRed[+] BcF6-GFP[-] cells are highlighted with a white spot and arrow. Both these nuclei are in the second row of nuclei from the surface of the lymph gland. (**a''**) At 12 h the cytoplasm around these nuclei is GFP[+]. (**B**) An independent sample of lymph gland ex vivo culture, with examples of BcF6-GFP[+] marked with green arrows. (**b'**) At 0 h, it is possible to identify 4 *Eater*-DsRed[+] BcF6-GFP[-] cells that will gain GFP expression. Two of these cells are in the second row of nuclei (arrows) and the other two are deeper in the tissue (white arrowheads). (**b''**) At 12 h the cytoplasm around these nuclei is GFP[+]. Scale bars represent 15 μm (**A**, **a'**, **a''**) and 10 μm (**B**, **B'**, **B''**).

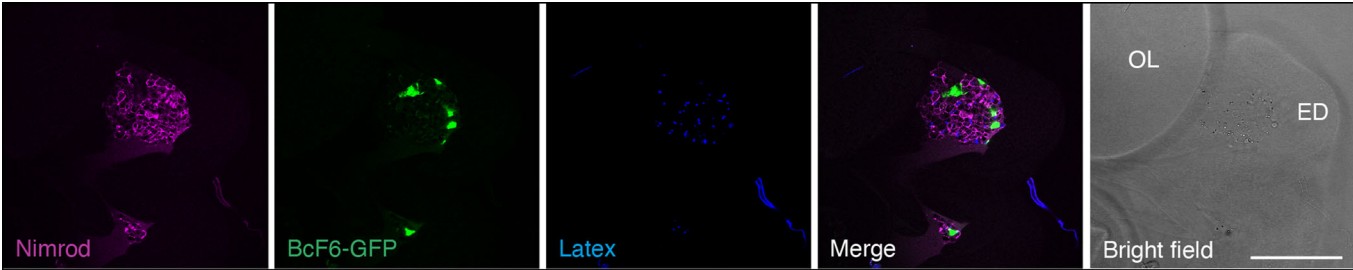

**Figure EV5. Latex beads are taken up exclusively by haemocytes.**

Latex beads co-localize only with the haemocyte patch (outlined in white) identified by the expression of Bc (crystal cells) or P1 (plasmatocytes). Cells of the neighboring eye disc (ED) and optic lobe (OL), visible in the merge image, show no phagocytosis of the latex beads. Scale bar represents 100 µm.

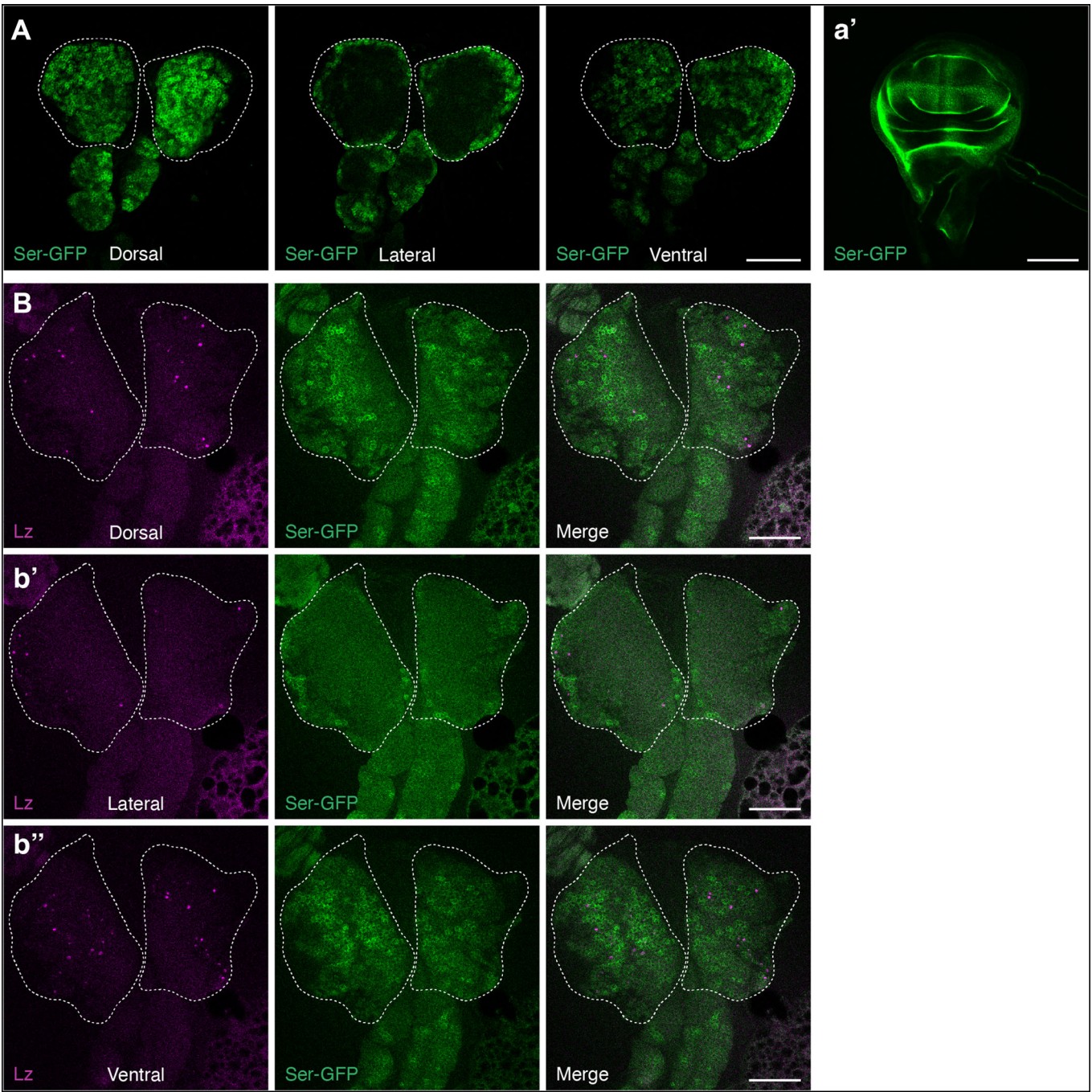

**Figure EV6. Serrate expression in the larval lymph gland.**

A Serrate eGFP protein fusion (Nagarkar-Jaiswal et al, 2015) appears restricted to a shallow layer of cortical plasmatocytes (**A**, **B**, **b'**, **b"**) and absent in the medullary prohaemocytes (**a'**, **b'**) of L3 larvae imaginal discs. This Ser-GFP fully recapitulates the endogenous expression in the wing disc when compared to published Serrate immunostainings (Lai, 2005) based on its pattern in the wing imaginal disc (**a'**). Crystal cell differentiation (revealed with anti-Lz) is limited to regions of Serrate expression (**B**, **b'**, **b"**). Dotted lines define the contour of the primary lymph gland lobes. Scale bars represent 100 μm.

