## [Peer Review File · EMBO Reports]

Transdifferentiation of plasmacytes to crystal cells in the lymph gland of *Drosophila melanogaster*

Julien Marcetteau, Patrícia Duarte, Alexandre Leitão, and Élio Sucena

Corresponding author(s): Élio Sucena (jesucena@fc.ul.pt) , Alexandre Leitão (alexandre.leitao@research.fchampalimaud.org)

Review Timeline:

Submission Date:	6th Feb 24
Editorial Decision:	28th Mar 24
Revision Received:	16th Oct 24
Editorial Decision:	12th Dec 24
Revision Received:	16th Dec 24
Accepted:	20th Dec 24

Editor: Deniz Senyilmaz Tiebe

Transaction Report:

Dear Dr. Sucena,

Thank you for the submission of your research manuscript to our journal, which was now seen by three referees, whose reports are copied below.

My apologies for this unusual delay in getting back to you. It took longer than anticipated to receive the full set of referee reports.

Referees express interest in the proposed transdifferentiation of plasmacytes to crystal cells. However, they also raise significant concerns that need to be addressed to consider publication here.

Given these positive recommendations, we would like to invite you to submit a revised manuscript. Please revise your manuscript with the understanding that the referee concerns (as in their reports) must be fully addressed and their suggestions taken on board. Please address all referee concerns in a complete point-by-point response. Acceptance of the manuscript will depend on a positive outcome of a second round of review. It is EMBO reports policy to allow a single round of major experimental revision only and acceptance or rejection of the manuscript will therefore depend on the completeness of your responses included in the next, final version of the manuscript.

We realize that it is difficult to revise to a specific deadline. In the interest of protecting the conceptual advance provided by the work, we recommend a revision within 3 months. Please discuss the revision progress ahead of this time with me if you require more time to complete the revisions, or if you have questions or comments regarding the revision (also by video chat).

1. A data availability section providing access to data deposited in public databases is missing (where applicable).
2. Your manuscript contains statistics and error bars based on $n=2$. Please use scatter plots in these cases.

You can submit the revision either as a Scientific Report or as a Research Article. For Scientific Reports, the revised manuscript can contain up to 5 main figures and 5 Expanded View figures, and it should not exceed 27000 characters. If the revision leads to a manuscript with more than 5 main figures it will be published as a Research Article. In this case the Results and Discussion section should be separate. If a Scientific Report is submitted, these sections have to be combined. This will help to shorten the manuscript text by eliminating some redundancy that is inevitable when discussing the same experiments twice. In either case, all materials and methods should be included in the main manuscript file.

4) a .docx formatted letter INCLUDING the reviewers' reports and your detailed point-by-point responses to their comments. As part of the EMBO publication's Transparent Editorial Process, EMBO reports publishes online a Review Process File (RPF) to accompany accepted manuscripts. This File will be published in conjunction with your paper and will include the referee reports, your point-by-point response and all pertinent correspondence relating to the manuscript.

<https://www.embopress.org/page/journal/14693178/authorguide#transparentprocess>

5) a complete author checklist, which you can download from our author guidelines <https://www.embopress.org/page/journal/14693178/authorguide>. Please insert information in the checklist that is also reflected in the manuscript. The completed author checklist will also be part of the RPF.

6) Please note that all corresponding authors are required to supply an ORCID ID for their name upon submission of a revised manuscript (<<https://orcid.org/>>). Please find instructions on how to link your ORCID ID to your account in our manuscript tracking system in our Author guidelines <<https://www.embopress.org/page/journal/14693178/authorguide#authorshipguidelines>>

Additional information on source data and instruction on how to label the files are available: <https://www.embopress.org/page/journal/14693178/authorguide#sourcedata>

9) Our journal encourages inclusion of *data citations in the reference list* to directly cite datasets that were re-used and obtained from public databases. Data citations in the article text are distinct from normal bibliographical citations and should directly link to the database records from which the data can be accessed. In the main text, data citations are formatted as follows: "Data ref: Smith et al, 2001" or "Data ref: NCBI Sequence Read Archive PRJNA342805, 2017". In the Reference list, data citations must be labeled with "[DATASET]". A data reference must provide the database name, accession number/identifiers and a resolvable link to the landing page from which the data can be accessed at the end of the reference. Further instructions are available at <http://www.embopress.org/page/journal/14693178/authorguide#referencesformat>

- the name of the statistical test used to generate error bars and P values,
- the number (n) of independent experiments (please specify technical or biological replicates) underlying each data point,
- the nature of the bars and error bars (s.d., s.e.m.),
- If the data are obtained from n Program fragment delivered error `Can't locate object method "less" via package "than" (perhaps you forgot to load "than"?) at //ejpvfs23/sites23b/embor_www/letters/embor_decision_revise_and_review.txt line 56.' 2, use scatter blots showing the individual data points.

12) Please also note our reference format:

I look forward to seeing a revised version of your manuscript when it is ready. Please let me know if you have questions or comments regarding the revision.

Kind regards,

Deniz Senyilmaz Tiebe

Deniz Senyilmaz Tiebe, PhD
Scientific Editor
EMBO Reports

Referee #1:

This study presents as an extension to the previously published study of the transdifferentiation of crystal cells in the sessile hemocyte cluster of *Drosophila* larvae. The video provides definitive proof of the recurrent phenomenon in the lymph gland. However, some of the details need to be clarified to prove the authors' model.

Although the authors nicely showed the presence of transdifferentiation of plasmacytes into crystal cells in the lymph gland, they did not clarify the lymph gland zone where this transdifferentiation occurs. Since most progenitor-derived blood cells are specified in the intermediate zone, this type of differentiation likely takes place exclusively in the cortical zone, not the intermediate zone (as in Figure 6). This point should be substantiated to conclude as described in Figure 6.

And related to the above question: The authors showed that Lz>Gtrace crystal cells did not co-localize with P1+ plasmacytes; however, it is not clear whether crystal cells derived from the intermediate zone (progenitor) are devoid of the mature plasmacyte marker.

Fig2a, Anti-P1 and anti-Lz cannot be used simultaneously. However, Lz>GFP may not be the most ideal genetic background to prove the authors' model as GFP may have a perdurance expression. Can the authors apply an alternative method to confirm this important experiment? For example, anti-Lz with Nimrod-GFP or eater>GFP would complement this experiment. Likewise, some of the important observations need to be validated with antibodies.

Can the transdifferentiated crystal cells express mature crystal cell markers, such as PPO1 or PPO2? It is also important to assess whether these transdifferentiating cells are destined to be functional in the lymph gland.

Fig2d, The authors confirmed that Notch RNAi driven by eater-Gal4 significantly reduced the number of crystal cells. Although eater is specific to mature plasmacytes, eater-GFP, which uses the same enhancer as eater-Gal4, co-localizes with Dome-Gal4 (Blanco-Obregon et al., 2020; Kroeger et al., 2011). If eater-Gal4 co-localizes with Dome, eater>N RNAi experiment cannot distinguish the effect of Notch in the intermediate zone/progenitor-derived crystal cells versus transdifferentiated crystal cells. An additional experiment is required to complement this caveat.

Line 187, correction of "as"

Line 189, correction of "carry"

Line 285, please check the sentence

Line 293, correction of "and"

Genotypes are not indicated in the figures. Please label the figures with corresponding genotypes.

Referee #2:

Summary:

The purpose of this study is to determine whether transdifferentiation plays a role in crystal cell formation in the lymph gland, like its role in sessile hemocyte clusters. To this end, I believe the author's data supports the general conclusion from the manuscript that transdifferentiation (from plasmacyte to crystal cell) occurs in the lymph gland.

While the data as a whole are convincing, at times, experimental data do not seem to correlate well with the specific conclusions made by the authors. The authors occasionally use imprecise language to describe phenomena or jump to seemingly disconnected conclusions. This in the end weakens their otherwise convincing data as it seems to stretch it beyond the limits of

what it shows. Nevertheless, I believe this notion of transdifferentiation from functionally mature plasmacytes into crystal cells in the lymph gland is an important conceptual advance.

Specific comments:

There are several ways that the clarity of the writing might be improved or that changes could be made to do the study justice without overselling the claims:

1. Hml-DsRed is not considered a marker of "mature" hemocytes (line 92) as it marks intermediate cells and "maturing hemocytes" (shown in many papers cited on p4). While the antibody P1 for Nimrod is considered a marker of functionally mature phagocytic plasmacytes. Overlap of Hml-DsRed with a crystal cell marker has been shown before and was interpreted as a "maturing" hemocyte becoming a crystal cell. The initial evidence here with Hml-DsRed supports that notion. But the P1 staining (which is a marker of functionally mature plasmacytes) with NRE-GFP, Lz-GFP and Su(H)-LacZ (Figures 1-2c) is much stronger evidence that a mature plasmacyte can become a crystal cell. The author's language should be clarified to make this distinction clear.

2. Line 103: "rare" does not equal "transient". The data showing a very small percentage of cells double positive for NRE and P1 shows it is a rare phenomenon but not that it is a transient state and does not "verify" (line 103) or "confirm" (line 115) the transient nature of the P1+/NRE+ cell type as stated by the authors on p5. "Transient" requires an observation of time. Furthermore, without some comparison of NRE-GFP to a true crystal cell marker (like Lz or BcF6) it is unclear whether NRE-GFP is just transiently expressed in any cell becoming a crystal cell or if the P1+/NRE+ cell type is a true transient state. In fact, as shown in supplementary figure 1, the presence of double positive P1+ and Lz>GFP or BcF6-GFP (which the author suggests are markers of committed later-stage crystal cells (p6)) suggests P1+ crystal cells may not be a strictly transient state in early crystal cells. And the fact that the authors state that in their live imaging they did not see Eater-DsRed shut-off (line 269-270) could suggest that the double positive state is a rare but final cell type/state and not a "transient" state.

3. Figure 2D:

a) The ** designation on Figure 2d is misleading as most conventions would represent ** as <0.01 and in this case it represents $p=0.02$ (conventionally *, <0.05). Instead of asterisks which are not explained in the figure legend, p-values should be written out on the graphs.

b) I am not a statistician, but it seems like a Wilcoxon (paired?) test may not be the correct statistical analysis for the data in 2d. A t-test with unpaired values seems more appropriate.

c) The authors use eater-gal4 as a marker of phagocytic plasmacytes (line 170) but their own lineage tracing data in supplementary figure 2 seems to suggest that eater expression is not restricted to the cortical zone and the authors also note that concerns have been raised regarding the specificity of the eater marker (line 186). A secondary driver for Notch RNAi in the cortical zone (and corresponding statistical tests) would strengthen their arguments and provide a better picture of the contribution of this "transdifferentiation" process to the overall crystal cell numbers in the lymph gland.

d) Alternatively, the authors could compare P1 staining with eater-gal4 expression to determine if all eater+ cells are in fact P1+ phagocytic plasmacytes or if eater expression marks an earlier less mature plasmacyte state. If eater marks both mature and "maturing" hemocytes, conclusions from Eater>NotchRNAi experiments would need to be reevaluated.

4. NRE-GFP seems to mark many fewer cells than other crystal cell markers like BcF6-GFP or Lz staining and so the authors conclusion that the lack of NRE-GFP expression in the medullary zone supports the idea that crystal cell differentiation is not occurring there (lines 94-96) does not seem well supported (especially because BcF6-GFP, another marker of crystal cells that seems to be present in many more cells) does seem to be in the medullary zone in supplementary figure 1b. Furthermore, later experiments showing a relatively weak decrease in crystal cell number after loss of Notch with eater-gal4 seems to suggest that transdifferentiation is a minor route to crystal cell formation, which begs the question of how else crystal cells form. The authors' conclusions that crystal cells do not come from the medullary zone and that they rarely come from transdifferentiation are confusing unless there is a third way of making a crystal cell the authors believe is the dominant mode. Furthermore, the authors own model in Figure 6 seems to show crystal cell formation at or near the medullary zone prohemocytes so it is hard to reconcile these data and conclusions.

5. The lineage tracing experiments with eater-LT and Lz staining support their hypothesis of transdifferentiation from eater+ to Lz+ but the authors state that no overlap is seen between LzLT and Nimrod staining. And yet, in the example image shown in Figure 3b, at least 3 of the LzLT+ cells seem to show Nimrod staining (albeit in the cytoplasm, not at the surface). And it seems like there is also a dim LzLT+ cell with P1 staining at the surface in the same image (upper right cell). Can the authors clarify this discrepancy? These data could support the authors' other model that a cell becomes double positive for both Lz and Nimrod and then downregulates Nimrod when it commits to the Lz fate. You would therefore expect a dimmer LzLT+ cell (earlier in its commitment) to have higher P1 expression at its surface and a later/brighter LzLT+ cell to have signs of P1 degradation (possibly cytoplasmic P1 staining). Quantitative analysis of the level of P1 staining in Lz-LT+ cells could clarify these observations and help better distinguish between their models.

6. Formatting issues (text and figures) and incorrect or incomplete labeling of figures or data distract and make it harder for the reader to fully evaluate the data:

a) My version of the manuscript had pervasive formatting issues with the letter E that I assume will be corrected in the final

manuscript (see lines 24, 33, 91, 92, 99, etc)

b) The legend in Figure 3a' is incomplete (blue and red are the same)

c) Only one of the white arrowheads in Figure 1A seems to point to double-positive plasmacytes with NRE-GFP

d) 1B: how many cells and lymph glands were quantified to arrive at these statistics?

e) The graph in figure 4 is misleading and does not seem to display the figures mentioned in the figure legend. Perhaps a different way of graphing these data or clearer annotation on the figure would help.

f) Different colored arrowheads should be used in Figure 5 to denote different cell populations of interest.

7. The statement on lines 299-301 that "genetic ablation of the posterior signaling center (PSC) has no impact on homeostatic blood cell types..." is incorrect for several reasons:

a) The references should be Benmimoun et al. 2015 and Oyallon et al. 2016 (not Benmimoun et al 2012)

b) Both Benmimoun et al. 2015 and Oyallon et al. 2016 show that genetic ablation of the PSC does in fact affect crystal cell numbers. In both cases, PSC loss decreases CC numbers (which both papers state in their results)

c) There is no clear connection between why PSC loss (and its effect or lack of an effect on CC numbers) would have anything to do with transdifferentiation or how transdifferentiation "could provide a mechanism that underlies this observation".

Overreaching and unclear statements like these may weaken the conclusions and confuse the reader.

Referee #3:

Drosophila has been studied extensively to investigate the regulation of insect blood cell (i.e., haemocytes) proliferation and differentiation. Different haematopoietic organs are formed during embryonic and larval development, and these contribute to various classes of specialized haemocytes during the larval (and adult) stage. The classic model is that undifferentiated prohaemocytes can differentiate through a binary switch, either into phagocytotic plasmacytes, or into crystal cells that produce pro-phenoloxidase. More recent research showed that in the more peripheral haematopoietic organs, transdifferentiation from plasmacytes to crystal cells occurs.

This current study is a rigorous investigation to show that transdifferentiation of mature plasmacytes into crystal cells also occurs in the major larval haematopoietic organ, the lymph gland. This transdifferentiation provides an additional mechanism for crystal cell production, complementing the canonical mechanisms of differentiation from prohaemocytes. Moreover, this transdifferentiation is not by cell division, but by direct transdifferentiation. This illustrates the pervasive plasticity in haemocytes, even those that had already fully functionally differentiated.

The combinations of experiments is strong, and the interpretation of the results is careful and rigorous. Alternative scenario's are being considered and described. The authors then convincingly show that the results are most compatible with the transdifferentiation scenario. These novel findings from this study are of high interest to molecular biologists, especially those working in developmental biology, but also those in regeneration biology, disease and homeostatic balance. They show that haematopoietic plasticity extends to beyond what was previously considered, and this *Drosophila* model may enable the further investigation of both the mechanistic regulation, and the function for having this additional mechanism.

The paper is very well written, providing extensive background on the current state of affairs/knowledge in the field. It is concise and clear, and the claims are well-justified and discussed with consideration.

1) My only suggestion to further improve this manuscript is a bit more extensive description on the functional aspects of the crystal cells: what are the primary functions of the pro-phenoloxidase producing crystal cells? And what are their dynamics under both homeostatic conditions (as in the current manuscript), and under more challenging conditions? The suggestion that this additional mechanism of transdifferentiation may especially relate to stress from hypoxia may also be discussed a bit more extensively.

Minor comments:

- Capital E in author and gene names did not convert well in the PDF version (questions marks, rather than the letter). Please, correct.

- In figure 3a, it is difficult to see whether the two fluorescence signals indeed come from a single (double positive) cell, or whether these could possibly also be from adjacent cells. This is easier in 3b.

- Line 187 (typo): as  has (haemocytes as shown)

- Line 191 (missing word): add "we" before "were"

- Line 380: abbreviation used for lymph gland (LG) - inconsistent

Dear Colleagues

Below we address each of your criticisms, suggestions and enquiries point by point. Your comments are in italics and our replies in plain text.

Thank you all for the constructive input that made the manuscript better. We hope to have addressed everything to your satisfaction.

Reviewer #1:

This study presents an extension to the previously published study of the transdifferentiation of crystal cells in the sessile hemocyte cluster of Drosophila larvae. The video provides definitive proof of the recurrent phenomenon in the lymph gland. However, some of the details need to be clarified to prove the authors' model.

Although the authors nicely showed the presence of transdifferentiation of plasmatocytes into crystal cells in the lymph gland, they did not clarify the lymph gland zone where this transdifferentiation occurs. Since most progenitor-derived blood cells are specified in the intermediate zone, this type of differentiation likely takes place exclusively in the cortical zone, not the intermediate zone (as in Figure 6). This point should be substantiated to conclude as described in Figure 6.

We thank the reviewer for this input. With the model in figure 6, our intention was to show transdifferentiation in the cortical zone, and clearly this was not achieved. We have updated the model to clarify that this mode of differentiation includes cortical and intermediate zones.

And related to the above question: The authors showed that Lz>Gtrace crystal cells did not co-localize with P1+ plasmatocytes; however, it is not clear whether crystal cells derived from the intermediate zone (progenitor) are devoid of the mature plasmatocyte marker.

Indeed, in our work, we do not test if progenitor derived crystal cells express *NimC1*. However, evidence from the literature suggests that the majority of crystal cells in the lymph gland follow a developmental path with no expression of *NimC1*. For example, in Girard et al, 2021, immature crystal cells iCC-5 differentiate from intermediary zone hemocytes IZ-5, without expression of *NimC1*. Only the small cluster of immature crystal cells iCC-7, that share a similar transcriptional profile with plasmacytes PL-7, have *NimC1* expression.

Fig2a, Anti-P1 and anti-lz cannot be used simultaneously. However, lz>GFP may not be the most ideal genetic background to prove the authors' model as GFP may have a perdurance expression. Can the authors apply an alternative method to confirm this important experiment? For example, anti-lz with Nimrod-GFP or eater>GFP would complement this experiment. Likewise, some of the important observations need to be validated with antibodies.

With this experiment we tested if a population of plasmacytes with a mature marker (NimC1) would express early markers of crystal cell fate. We agree that GFP may have a perduring expression in this experiment. However, if these double positive cells are not NimC1+ cells starting to express *Iz*, then the alternative hypothesis is that these cells eventually lose *Iz* expression and become plasmacytes (NimC1+). We find no evidence of this process as shown in Fig3b where lineage tracing of *Iz*+ cells never give rise to Nimrod+ cells.

Can the transdifferentiated crystal cells express mature crystal cell markers, such as PPO1 or PPO2? It is also important to assess whether these transdifferentiating cells are destined to be functional in the lymph gland.

Unfortunately, we have limited evidence to offer. However, we can contend that, as shown in In Fig2b, Nimrod+ cells express Bc-GFP, i.e., cells are using the promoter of *PPO1* gene. And these double positive cells are the very small population of cells that our results indicate are transdifferentiating into crystal cells. Hence, based on this evidence we believe them to be functional crystal cells.

Fig2d. The authors confirmed that Notch RNAi driven by eater-Gal4 significantly reduced the number of crystal cells. Although eater is specific to mature plasmacytes, eater-GFP, which uses the same enhancer as eater-Gal4, co-localizes with Dome-Gal4 (Blanco-Obregon et al., 2020; Kroeger et al., 2011). If eater-Gal4 co-localizes with Dome, eater>N RNAi experiment cannot distinguish the effect of Notch in the intermediate zone/progenitor-derived crystal cells versus transdifferentiated crystal cells. An additional experiment is required to complement this caveat.

We thank the reviewer for this comment, which highlights a problem very difficult to deal with in this study. An ideal experiment to address this problem is to use a GAL4 line that specifically labels mature plasmacytes in the cortical zone. We tried to look for such a line in the literature. We found no reference describing a driver line that is expressed in cortical plasmacytes and has no expression in the medullary zone and no expression in cortical crystal cells.

The most promising course of action was to explore a set of lines described in Tokusumi et al. (G3, 2017). In this work, five GAL4 drivers were described as having expression in plasmacytes but not in crystal cells, and restricted expression to the cortical zone (Bloomington#: 45737; 49466; 45304; 47935; 48154). We tested all five lines but could not confirm these expression patterns. In three lines (45737; 49466; 48154) we could not observe the described expression in haemocytes, although we see expression in other tissues. In two lines, that have different regions of *pointed* promoter driving GAL4 (45304; 47935), we see expression in the cortical zone of the lymph gland, but both plasmacytes and crystal cells are marked.

We agree that this problem needs to be addressed for future studies on haematopoiesis in the lymph gland. But we envision this to be a hard goal to

achieve. For example, in Cho et al, (NatCom, 2020) where an extensive analysis of scRNA-seq was performed in lymph gland haemocytes, it is possible to find specific markers of lamellocytes and crystal cells, i.e., genes that have exclusive expression in those cell types in the lymph gland. No such marker seems to exist for plasmatocytes. For example, NimC1 still has considerable expression in crystal cells cluster 1 (CC1).

Even if there is no gene that is specific to plasmatocytes, it may be the case that mature plasmatocytes use specific regulatory regions. Hopefully, in future screens with different promotor regions may find a specific driver for cortical plasmatocytes.

With this caveat in our experiment, we made sure to confirm that the transdifferentiation process is taking place in the cortical zone. For that, we reanalyzed the images from our experiments with lymph gland *ex vivo* cultures. Specifically, we looked for examples of cells in outer part of the lymph gland (cortical zone) that gain GFP expression driven by Bc-GFP. By performing spot segmentation and tracking in IMARIS, we were able to identify the initial position of cells that gain GFP expression throughout the video. In two lymph glands we were able to identify 4 cells located in the second layer that gain this expression.

We have changed the text in the results part to address this problem explicitly:

Line 205: “Morphological analysis of these samples with segmented and tracked nuclei allowed us to determine where new crystal cells are differentiating in the lymph gland (Supplementary figure 4). We could confirm the presence of transdifferentiating cells within the outer 2 cell layers, a region that corresponds to the cortical zone. This observation is crucial because *eater-dsRed*, which uses the same enhancer as *eater-GAL4*, is expressed in distal progenitors of the intermediary zone (Blanco-Obregon et al. 2020). Consequently, our observations from the Notch knockdown experiment (Figure 2d) and the analysis of *eater* lineage tracing (Figure 3) could be explained by events occurring solely the distal part of the intermediary zone.”

- *Line 187, correction of "as"*
Changed for “has”
- *Line 189, correction of "carry"*
Changed to “conduct”
- *Line 285, please check the sentence*
Changed to “For example, this strategy may be applied to questions about the dynamics of cell differentiation over time and space, where live imaging is essential.”
- *Line 293, correction of "and"*
Changed for “an”
- *Genotypes are not indicated in the figures. Please label the figures with corresponding genotypes.*

We now indicate the genotypes in the figure legend(s).

Reviewer #2:

Summary:

The purpose of this study is to determine whether transdifferentiation plays a role in crystal cell formation in the lymph gland, like its role in sessile hemocyte clusters. To this end, I believe the author's data supports the general conclusion from the manuscript that transdifferentiation (from plasmatocyte to crystal cell) occurs in the lymph gland.

While the data as a whole are convincing, at times, experimental data do not seem to correlate well with the specific conclusions made by the authors. The authors occasionally use imprecise language to describe phenomena or jump to seemingly disconnected conclusions. This in the end weakens their otherwise convincing data as it seems to stretch it beyond the limits of what it shows. Nevertheless, I believe this notion of transdifferentiation from functionally mature plasmatocytes into crystal cells in the lymph gland is an important conceptual advance.

Specific comments:

There are several ways that the clarity of the writing might be improved or that changes could be made to do the study justice without overselling the claims:

1. Hml-DsRed is not considered a marker of "mature" hemocytes (line 92) as it marks intermediate cells and "maturing hemocytes" (shown in many papers cited on p4). While the antibody P1 for Nimrod is considered a marker of functionally mature phagocytic plasmatocytes. Overlap of Hml-DsRed with a crystal cell marker has been shown before and was interpreted as a "maturing" hemocyte becoming a crystal cell. The initial evidence here with Hml-DsRed supports that notion. But the P1 staining (which is a marker of functionally mature plasmatocytes) with NRE-GFP, Lz-GFP and Su(H)-LacZ (Figures 1-2c) is much stronger evidence that a mature plasmatocyte can become a crystal cell. The author's language should be clarified to make this distinction clear.

Thank you for noticing this inaccuracy. We revised the paragraph to have a more precise description:

Line 92: "Firstly, combining NRE-eGFP with Hml Δ -DsRed (a cell marker that starts its expression in maturing haemocytes) and observing 12 dissected lymph glands could not reveal NRE-eGFP expression in the medullary zone, which hosts undifferentiated haemocytes (Supplementary Figure 1a and a'). The restricted expression pattern of Notch activity to the intermediate and cortical zone of Hml+ haemocytes is consistent with crystal cell differentiation not occurring in the medullary zone.

2. Line 103: "rare" does not equal "transient". The data showing a very small percentage of cells double positive for NRE and P1 shows it is a rare phenomenon but not that it is a transient state and does not "verify" (line 103) or "confirm" (line 115) the transient nature of the P1+/NRE+ cell type as stated by the authors on p5. "Transient" requires an observation of time. Furthermore, without some comparison of NRE-GFP to a true crystal cell marker (like Lz or

BcF6) it is unclear whether NRE-GFP is just transiently expressed in any cell becoming a crystal cell or if the P1+/NRE+ cell type is a true transient state. In fact, as shown in supplementary figure 1, the presence of double positive P1+ and Lz>GFP or BcF6-GFP (which the author suggests are markers of committed later-stage crystal cells (p6)) suggests P1+ crystal cells may not be a strictly transient state in early crystal cells. And the fact that the authors state that in their live imaging they did not see Eater-DsRed shut-off (line 269-270) could suggest that the double positive state is a rare but final cell type/state and not a "transient" state.

We agree with the reviewer that, based solely in our results, the double positive cells might be interpreted as a final state of a rare cell type. However, we point in the discussion that our interpretation of the results is guided by the recent sc-RNA studies in the lymph gland:

Line 241: "This model is supported further by recent transcriptomics studies, showing crystal cells differentiation from hemocyte clusters expressing mature plasmatocyte markers, such as Eater and NimC1. During crystal cell differentiation, these markers are downregulated while mature crystal cell markers are upregulated (Cho et al., 2020; Girard et al., 2021)."

For example, in Girard et al, 2021, immature crystal cells iCC-7 share a transcriptomic profile with plasmatocytes PL-7. Both these cell clusters express *NimC1* and *Iz*, but following their development to iCC-6 the expression of *NimC1* is reduced and *Iz* increased.

We believe this to be the most parsimonious explanation of our results, given all the evidence found in the literature.

3. Figure 2D:

a) The ** designation on Figure 2d is misleading as most conventions would represent ** as <0.01 and in this case it represents $p=0.02$ (conventionally *; <0.05). Instead of asterisks which are not explained in the figure legend, p-values should be written out on the graphs.

The asterisks were replaced with the the p-value in this figure

b) I am not a statistician, but it seems like a Wilcox (paired?) test may not be the correct statistical analysis for the data in 2d. A t-test with unpaired values seems more appropriate.

A Wilcox test was chosen here because the data failed a normality test.

c) The authors use *eater-gal4* as a marker of phagocytic plasmatocytes (line 170) but their own lineage tracing data in supplementary figure 2 seems to suggest that eater expression is not restricted to the cortical zone and the authors also note that concerns have been raised regarding the specificity of the eater marker (line 186). A secondary driver for Notch RNAi in the cortical zone (and corresponding statistical tests) would strengthen their arguments and

provide a better picture of the contribution of this "transdifferentiation" process to the overall crystal cell numbers in the lymph gland. & d) Alternatively, the authors could compare P1 staining with eater-gal4 expression to determine if all eater+ cells are in fact P1+ phagocytic plasmatocytes or if eater expression marks an earlier less mature plasmatocyte state. If eater marks both mature and "maturing" hemocytes, conclusions from Eater>NotchRNAi experiments would need to be reevaluated.

We thank the reviewer for raising this important issue. This was also raised by reviewer 1, so we replicate our answer here:

This comment highlights a problem very difficult to deal with in this study. An ideal experiment to address this problem is to use a GAL4 line that specifically labels mature plasmatocytes in the cortical zone. We tried to look for such a line in the literature. We found no reference describing a driver line that is expressed in cortical plasmatocytes and has no expression in the medullary zone and no expression in cortical crystal cells.

The most promising course of action was to explore a set of lines described in Tokusumi et al. (G3, 2017). In this work, five GAL4 drivers were described as having expression in plasmatocytes but not in crystal cells, and restricted expression to the cortical zone (Bloomington#: 45737; 49466; 45304; 47935; 48154). We tested all five lines but could not confirm these expression patterns. In three lines (45737; 49466; 48154) we could not observe the described expression in haemocytes, although we see expression in other tissues. In two lines, that have different regions of *pointed* promotor driving GAL4 (45304; 47935), we see expression in the cortical zone of the lymph gland, but both plasmatocytes and crystal cells are marked.

We agree that this problem needs to be addressed for future studies on haematopoiesis in the lymph gland. But we envision this to be a hard goal to achieve. For example, in Cho et al, (NatCom, 2020) where an extensive analysis of scRNA-seq was performed in lymph gland haemocytes, it is possible to find specific markers of lamellocytes and crystal cells, i.e., genes that have exclusive expression in those cell types in the lymph gland. No such marker seems to exist for plasmatocytes. For example, NimC1 still has considerable expression in crystal cells cluster 1 (CC1).

Even if there is no gene that is specific to plasmatocytes, it may be the case that mature plasmatocytes use specific regulatory regions. Hopefully, in future screens with different promotor regions may find a specific driver for cortical plasmatocytes.

With this caveat in our experiment, we made sure to confirm that the transdifferentiation process is taking place in the cortical zone. For that, we reanalyzed the images from our experiments with lymph gland *ex vivo* cultures. Specifically, we looked for examples of cells in outer part of the lymph gland (cortical zone) that gain GFP expression driven by Bc-GFP. By performing spot segmentation and tracking in IMARIS, we were able to identify the initial position of cells that gain GFP expression throughout the video. In two lymph

glands we were able to identify 4 cells located in the second layer that gain this expression.

We have changed the text in the results part to address this problem explicitly:

Line 205 : “Morphological analysis of these samples with segmented and tracked nuclei allowed us to determine where new crystal cells are differentiating in the lymph gland (Supplementary figure 4). We could confirm the presence of transdifferentiating cells within the outer 2 cell layers, a region that corresponds to the cortical zone. This observation is crucial because *eater-dsRed*, which uses the same enhancer as *eater-GAL4*, is expressed in distal progenitors of the intermediary zone (Blanco-Obregon et al. 2020). Consequently, our observations from the Notch knockdown experiment (Figure 2d) and the analysis of *eater* lineage tracing (Figure 3) could be explained by events occurring solely the distal part of the intermediary zone.”

4. NRE-GFP seems to mark many fewer cells than other crystal cell markers like BcF6-GFP or Iz staining and so the authors conclusion that the lack of NRE-GFP expression in the medullary zone supports the idea that crystal cell differentiation is not occurring there (lines 94-96) does not seem well supported (especially because BcF6-GFP, another marker of crystal cells that seems to be present in many more cells) does seem to be in the medullary zone in supplementary figure 1b. Furthermore, later experiments showing a relatively weak decrease in crystal cell number after loss of Notch with eater-gal4 seems to suggest that transdifferentiation is a minor route to crystal cell formation, which begs the question of how else crystal cells form. The authors' conclusions that crystal cells do not come from the medullary zone and that they rarely come from transdifferentiation are confusing unless there is a third way of making a crystal cell the authors believe is the dominant mode. Furthermore, the authors own model in Figure 6 seems to show crystal cell formation at or near the medullary zone prohemocytes so it is hard to reconcile these data and conclusions.

Yes! We studied crystal cell transdifferentiation in the lymph gland, without ever doubting that the canonical mechanism described for crystal cell development is the most prevalent. This was so engrained in our minds that we did not pass it to the text of the manuscript with sufficient clarity. We have changed the opening sentence of the discussion to pass that message clearly:

Line 244: “The overwhelmingly prevalent mode of crystal cell differentiation in the lymph gland is through a binary choice that occurs in immature haemocytes (progenitors), in a *Notch* dependent manner (reviewed in Banerjee *et al.* 2019).”

In addition, we have updated the model in figure 6, to make it clear that transdifferentiation occurs in the cortical zone and the canonical differentiation mechanism occurs in the medullary/intermediary zone of the lymph gland.

5. The lineage tracing experiments with *eater-LT* and *Lz* staining support their hypothesis of transdifferentiation from *eater+* to *Lz+* but the authors state that no overlap is seen between *LzLT* and *Nimrod* staining. And yet, in the example image shown in Figure 3b, at least 3 of the *LzLT+* cells seem to show *Nimrod* staining (albeit in the cytoplasm, not at the surface). And it seems like there is also a dim *LzLT+* cell with *P1* staining at the surface in the same image (upper right cell). Can the authors clarify this discrepancy? These data could support the authors' other model that a cell becomes double positive for both *Lz* and *Nimrod* and then downregulates *Nimrod* when it commits to the *Lz* fate. You would therefore expect a dimmer *LzLT+* cell (earlier in its commitment) to have higher *P1* expression at its surface and a later/brighter *LzLT+* cell to have signs of *P1* degradation (possibly cytoplasmic *P1* staining). Quantitative analysis of the level of *P1* staining in *Lz-LT+* cells could clarify these observations and help better distinguish between their models.

Again, we thank the reviewer for raising this issue. The apparent *P1* signal in *LzLT+* cells is, very likely, "bleed through" from the GFP channel. Notice that the lineage tracing marks the nuclei (*Stinger*), not the cytoplasm. The *P1* channel "signal" colocalizes with the GFP signal. This "bleed through" is a problem that we had in other images. Hence, in such cases, we took in consideration the localization of the signal.

6. Formatting issues (text and figures) and incorrect or incomplete labeling of figures or data distract and make it harder for the reader to fully evaluate the data:

a) My version of the manuscript had pervasive formatting issues with the letter *E* that I assume will be corrected in the final manuscript (see lines 24, 33, 91, 92, 99, etc).

This must have been a formatting error in the PDF conversion. We will try to fix this problem in the resubmission.

b) The legend in Figure 3a' is incomplete (blue and red are the same)

This has been corrected

c) Only one of the white arrowheads in Figure 1A seems to point to double-positive plasmacytes with *NRE-GFP*

The white arrows were chosen based on the raw images, which is not very clear in the manuscript figure, as pointed out. We have removed the less obvious cases to avoid confusion.

d) 1B: how many cells and lymph glands were quantified to arrive at these statistics?

This was obtained from 761 cells from 4 lymph glands. We have added the following to the figure legend:

"n = 761 cells from 4 lymph glands"

e) The graph in figure 4 is misleading and does not seem to display the figures mentioned in the figure legend. Perhaps a different way of graphing these data or clearer annotation on the figure would help.

Yes, this is of little help, we agree. Therefore, and because the numbers are detailed in the text, we have removed this graph.

f) Different colored arrowheads should be used in Figure 5 to denote different cell populations of interest.

The arrowhead showing the example of Bc-GFP+Nimc1+ cell with latex bead was changed to green.

7. The statement on lines 299-301 that "genetic ablation of the posterior signaling center (PSC) has no impact on homeostatic blood cell types..." is incorrect for several reasons:

a) The references should be Benmimoun et al. 2015 and Oyallon et al. 2016 (not Benmimoun et al 2012)

b) Both Benmimoun et al. 2015 and Oyallon et al. 2016 show that genetic ablation of the PSC does in fact affect crystal cell numbers. In both cases, PSC loss decreases CC numbers (which both papers state in their results)

c) There is no clear connection between why PSC loss (and its effect or lack of an effect on CC numbers) would have anything to do with transdifferentiation or how transdifferentiation "could provide a mechanism that underlies this observation". Overreaching and unclear statements like these may weaken the conclusions and confuse the reader.

We thank the reviewer for making this correction, we agree that this sentence was overreaching our conclusions. What we meant was that the fact crystal cells are still observed in the absence of the PSC, may have to do with the process of transdifferentiation in the cortical zone. But we have no data to support this, so we have removed this sentence from the discussion section.

Reviewer #3:

Drosophila has been studied extensively to investigate the regulation of insect blood cell (i.e., haemocytes) proliferation and differentiation. Different haematopoietic organs are formed during embryonic and larval development, and these contribute to various classes of specialized haemocytes during the larval (and adult) stage. The classic model is that undifferentiated prohaemocytes can differentiate through a binary switch, either into phagocytotic plasmatocytes, or into crystal cells that produce pro-phenoloxidase. More recent research showed that in the more peripheral haematopoietic organs, transdifferentiation from plasmatocytes to crystal cells occurs.

This current study is a rigorous investigation to show that transdifferentiation of mature plasmatocytes into crystal cells also occurs in the major larval haematopoietic organ, the lymph gland. This transdifferentiation provides an additional mechanism for crystal cell production, complementing the canonical mechanisms of differentiation from prohaemocytes. Moreover, this transdifferentiation is not by cell division, but by direct transdifferentiation. This illustrates the pervasive plasticity in haemocytes, even those that had already fully functionally differentiated.

The combinations of experiments is strong, and the interpretation of the results is careful and rigorous. Alternative scenarios are being considered and described. The authors then convincingly show that the results are most compatible with the transdifferentiation scenario. These novel findings from this study are of high interest to molecular biologists, especially those working in developmental biology, but also those in regeneration biology, disease and homeostatic balance. They show that haematopoietic plasticity extends to beyond what was previously considered, and this *Drosophila* model may enable the further investigation of both the mechanistic regulation, and the function for having this additional mechanism.

The paper is very well written, providing extensive background on the current state of affairs/knowledge in the field. It is concise and clear, and the claims are well-justified and discussed with consideration.

1) My only suggestion to further improve this manuscript is a bit more extensive description on the functional aspects of the crystal cells: what are the primary functions of the pro-phenoloxidase producing crystal cells? And what are their dynamics under both homeostatic conditions (as in the current manuscript), and under more challenging conditions? The suggestion that this additional mechanism of transdifferentiation may especially relate to stress from hypoxia may also be discussed a bit more extensively.

We agree these are really good points to tighten the discussion around crystal cells by expanding and exploring their function. However, we also wish to keep the other lines of discussion we have explored without extending the manuscript too much. So, we have opted to include a more explicit, and somewhat historical, account of crystal cell function whilst reiterating the

potential of our results in contributing to novel functional insights adding to this tradition. We have developed the section between lines 308-320 to include:

“In addition, in the absence of PPO2, larvae display a physiological state of hypoxia even under normoxic conditions as this product of crystal cells fails to ensure its role in haemocyte guidance and systemic oxygen circulation (Shin et al., 2024). In this light, it will be interesting to re-visit the relative weight of transdifferentiation under hypoxic conditions, and whether this mechanism may be called upon with higher prevalence under stress. Future enquiries will determine whether transdifferentiation of plasmatocytes into crystal cells is a general mechanism, under local or systemic stress, to increase the deployment and response capacity of these cells (Binggeli et al., 2014; Dudzic et al., 2015; Vlisidou & Wood, 2015; Letourneau et al., 2016; Cho et al., 2018; Ramesh et al., 2021). Such future enquiry holds the promise of extending fundamental knowledge, gathered over more than half a century, on crystal cell differentiation and function in response to external aggression to promote melanization and wound-healing (Rizki & Rizki, 1959; Rizki, & Rizki, 1980b; De Gregorio et al., 2002; Sorrentino, et al., 2002; Bidla et al., 2007; Eleftherianos & Revenis, 2011; Dudzic et al., 2015; Banerjee et al., 2019; Shin et al., 2024).”

We wish it could be longer too but hope this reaches an acceptable compromise with what the reviewer had in mind.

Minor comments:

- *Capital E in author and gene names did not convert well in the PDF version (questions marks, rather than the letter). Please, correct.*

This must have been a formatting error in the PDF conversion. We will fix this problem in the resubmission.

- *In figure 3a, it is difficult to see whether the two fluorescence signals indeed come from a single (double positive) cell, or whether these could possibly also be from adjacent cells. This is easier in 3b.*

An inset has been added to make it clearer.

- *Line 187 (typo): as  has (haemocytes as shown)*
Changed to “has”

- *Line 191 (missing word): add "we" before "were"*
Added “we”

- *Line 380: abbreviation used for lymph gland (LG) – inconsistent*
Changed to “lymph gland”.

Dear Élio,

Thank you for submitting your revised manuscript. It has now been seen by two of the original referees. Please accept my apologies for this unusual delay. As mentioned before, it took longer than anticipated to receive the referee reports.

As you can see, both referees find that the study is significantly improved during revision and recommend publication. However, I need you to address the points below before I can accept the manuscript.

- Please add a point in Discussion about the future directions for the study on distinguishing the developmental pathways of crystal cells (as per referee #1).
- Please address the remaining minor concerns of referee #2.
- Please place the keywords after the Abstract.
- Please add "Disclosure Statement and Competing Interests" as a title before the following sentence "The Authors declare they do not have competing interests."
- Please remove the Author Contributions section from the manuscript.
- As per our format requirements, in the reference list, citations should be listed in alphabetical order and then chronologically, with the authors' surnames and initials inverted; where there are more than 10 authors on a paper, 10 will be listed, followed by 'et al.'. Please see <https://www.embopress.org/page/journal/14693178/authorguide#referencesformat>
- Please update the citation format of preprint: Brooks et al, 2023 as follows:
In-text citation: (preprint: NAME1 et al, YEAR)
Reference list: Author NAME1, Author NAME2, (YEAR) article title. bioRxiv doi: 1234/002.dfj123 [PREPRINT]
- We note that the following information regarding funding is missing from the manuscript tracking system: "co-financed by Lisboa Regional Operational Programme (Lisboa 2020), under the Portugal 2020 Partnership Agreement, through the European Regional Development Fund (ERDF), and Foundation for Science and Technology (Portugal)."
- The correct nomenclature for Supplementary figures is Figure EV1, EV2 etc (stands for Expanded View Figure). Please update their source file names, titles in the manuscript tracking system, figure legends in the manuscript, callouts in the manuscript. (Please see <https://www.embopress.org/page/journal/14693178/authorguide#expandedview>).
- Related to the point above, please resubmit all main figures and EV figures as separate image files (one file per figure).
- The correct nomenclature for movies is Movie EV1. Its legend should be removed from the manuscript and provided in a readme.txt file that will then be zipped together and uploaded as Movie EV1 folder.
- All research articles submitted as revised versions must include a structured methods section that includes a Reagents and Tools Table followed by a Methods and Protocols section. Please see <https://www.embopress.org/page/journal/14693178/authorguide#structuredmethods> for further information.
- At EMBO Press, we ask the authors to provide the source data that were used to generate the main figures. Please submit source data as requested by our Source Data Coordinator Dr. Hannah Sonntag per email dated 02.04.2024. I pasted her email below and attached the source data checklist to be filled in and submitted as well.
- Our production/data editors have asked you to clarify several points in the figure legends:
 - o Please note that the legend for figure 1A, A' are not bifurcated in the manuscript. This needs to be rectified.
 - o Please note that the legend for supplementary figure 2A, A', A' are not bifurcated in the manuscript. This needs to be rectified."
 - o Please define the annotated p values ****/*/*/*/* as well as provide the statistical test used and the exact p-values for the same in the legend of figure 1C as appropriate.
 - o Please indicate the statistical test used for data analysis in the legends of figures 2D"
 - o Please note that the box plots need to be defined in terms of minima, maxima, centre, bounds of box and whiskers, and percentile in the legends of figures 1C, 2D
 - o Please note that information related to n is missing in the legends of figures 1C, 2D "
 - o Please note that the dotted line is not defined in the legend of figures 1A, 2A, B; 3A, 4B, B'; Supplementary figure(s) 1A, A', B, C; 2A, 5, 6A, A', B, B', B'. This needs to be rectified.
 - o Please note that the green arrow heads are not defined in the legend of figures 4B, B'. This needs to be rectified.
 - o Please note that legend of figure 2B references "arrowheads", however there are no arrowheads present in the figure. Kindly rectify the same.
 - o Please note that the white dotted box and white arrow are not defined in the legend of supplementary figure - This needs to be rectified."
- Papers published in EMBO Reports include a 'synopsis' and 'bullet points' to further enhance discoverability. Both are displayed on the html version of the paper and are freely accessible to all readers. The synopsis includes a short standfirst summarizing the study in 1 or 2 sentences (max 35 words) that summarize the paper and are provided by the authors and streamlined by the handling editor. I would therefore ask you to include your synopsis blurb and 3-5 bullet points listing the key experimental findings.
- In addition, please provide an image for the synopsis. This image should provide a rapid overview of the question addressed in the study but still needs to be kept fairly modest since the image size cannot exceed 550 (width) x 300-600 (height) pixels.

Thank you again for giving us to consider your manuscript for EMBO Reports, I look forward to your minor revision.

Kind regards,

Deniz

--

Deniz Senyilmaz Tiebe, PhD
Senior Scientific Editor
EMBO Reports

Referee #1:

The authors have adequately addressed all the critiques and revised their model in Figure 6. However, I believe that distinguishing the developmental pathways of crystal cells - whether through progenitor-derived differentiation or transdifferentiation - requires additional mechanistic evidence to fully support the authors' hypothesis. I hope future studies will explore the details of the transdifferentiation process in lymph gland crystal cells.

Referee #2:

The authors have adequately addressed my concerns. The additions and changes to the manuscript strengthen and enhance it. Overall this study is an important advancement in the field.

Typos on lines 73?, 414-417, 421,

For reproducibility, item numbers should be included for commercial reagents: primary antibodies for GFP and Lz, fluorescent secondaries, and latex beads.

Dear Colleagues

Below we address requests for modifications and editions, point by point. Your comments are in italics and our replies in plain text.

Thank you all for your thorough and constructive work. We hope to have addressed everything to your satisfaction.

As per the reviewers comments:

Referee #1:

The authors have adequately all the critiques and revised their model in Figure 6. However, I believe that distinguishing the developmental pathways of crystal cells - whether through progenitor-derived differentiation or transdifferentiation - requires additional mechanistic evidence to fully support the authors' hypothesis. I hope future studies will explore the details of the transdifferentiation process in lymph gland crystal cells.

We have added the following sentence to the discussion (lines:)

Referee #2:

The authors have adequately addressed my concerns. The additions and changes to the manuscript strengthen and enhance it. Overall this study is an important advancement in the field.

Typos on lines 73?, 414-417, 421,

For reproducibility, item numbers should be included for commercial reagents: primary antibodies for GFP and Lz, fluorescent secondaries, and latex beads.

We have corrected the typos and added a reagent table that addresses the reviewer's request.

As per editing:

1) We have corrected legends 1 and 2 as required:

- Include the necessary bifurcations
- mention the statistical tests performed
- Define the required parameters for box plots
- Include sample sizes
- Include arrowheads mentioned in the legend
- Clarify the p-value and its representation
- Removed dotted line from figures

2) We have corrected all references both on the reference list and throughout the text according to EMBO instructions.

3) We have added a Reagents and Tools Table the Methods and Protocols section.

4) we have re-ordered, remove or added the required sections from the main text.

5) we have corrected the nomenclature for figures, supplementary figures and movie, to comply to EMBO format and re-submitted each separately.

6) We have contacted Dr Hannah Sonntag with the source files and forms required. We hope to have it all in order.

At your disposal for any necessary further corrections.

All the best,
Élio Sucena
(on behalf of all authors)

Dr. Élio Sucena
Faculdade de Ciências da Universidade de Lisboa
Departamento de Biologia Animal
Edifício C2
Campo Grande
Lisboa 1749-016
Portugal

Dear Élio,

Thank you for submitting your revised manuscript. I have now looked at everything and all is fine. Therefore, I am very pleased to accept your manuscript for publication in EMBO Reports.

Congratulations on a nice work!

I need your input on more point before we can transfer your manuscript to our publishers. We note that the source data checklist has not been filled in (attached). You can return the filled in file by responding to this email. Thank you.

Kind regards,

Deniz

--

Deniz Senyilmaz Tiebe, PhD
Senior Scientific Editor
EMBO Reports

—
Your manuscript will be processed for publication by EMBO Press. It will be copy edited and you will receive page proofs prior to publication. Please note that you will be contacted by Springer Nature Author Services to complete licensing and payment information.
